# Ca$^{2+}$-activated sphingomyelin scrambling and turnover mediate ESCRT-independent lysosomal repair

Patrick Niekamp[1], Felix Scharte [2], Tolulope Sokoya[1], Laura Vittadello [3], Yeongho Kim [4], Yongqiang Deng[4], Elisabeth Südhoff[1], Angelika Hilderink[1], Mirco Imlau[3], Christopher J. Clarke [5], Michael Hensel[2], Christopher G. Burd[4] & Joost C. M. Holthuis [1✉]

Lysosomes are vital organelles vulnerable to injuries from diverse materials. Failure to repair or sequester damaged lysosomes poses a threat to cell viability. Here we report that cells exploit a sphingomyelin-based lysosomal repair pathway that operates independently of ESCRT to reverse potentially lethal membrane damage. Various conditions perturbing organelle integrity trigger a rapid calcium-activated scrambling and cytosolic exposure of sphingomyelin. Subsequent metabolic conversion of sphingomyelin by neutral sphingomyelinases on the cytosolic surface of injured lysosomes promotes their repair, also when ESCRT function is compromised. Conversely, blocking turnover of cytosolic sphingomyelin renders cells more sensitive to lysosome-damaging drugs. Our data indicate that calcium-activated scramblases, sphingomyelin, and neutral sphingomyelinases are core components of a previously unrecognized membrane restoration pathway by which cells preserve the functional integrity of lysosomes.

[1] Molecular Cell Biology Division, Department of Biology and Center of Cellular Nanoanalytics, University of Osnabrück, 49076 Osnabrück, Germany. [2] Microbiology Division, Department of Biology and Center of Cellular Nanoanalytics, University of Osnabrück, 49076 Osnabrück, Germany. [3] Experimental Physics Division, Department of Physics and Center of Cellular Nanoanalytics, University of Osnabrück, 49076 Osnabrück, Germany. [4] Department of Cell Biology, Yale School of Medicine, New Haven, CT 06520, USA. [5] Department of Medicine and Cancer Center, Stony Brook University, Stony Brook, NY 11794, USA. ✉email: holthuis@uos.de

Lysosomes are essential cellular organelles involved in the degradation of macromolecules, pathogen killing, and metabolic signaling. To perform these vital tasks, lysosomes contain high concentrations of acid hydrolases, protons, and calcium. Conversely, lysosomal damage caused by incoming pathogens, amphiphilic drugs, or sharp crystals can have deleterious consequences, including cell death[1]. To avoid the spilling of harmful lysosomal contents into the cytosol, injured lysosomes are marked for degradation by a specialized form of autophagy, known as lysophagy. This process is initiated by recruitment of cytosolic galectins and glycoprotein-specific ubiquitin ligases to abnormally exposed luminal glycans at the lesion site, resulting in engulfment of the damaged lysosome by autophagic membranes[2,3]. While lysophagy is a slow process in which the disrupted organelle is ultimately sacrificed, recent work revealed an important role of the Endosomal Sorting Complex Required for Transport (ESCRT) machinery in repairing small perforations in lysosomes to allow their escape from autophagic degradation[4,5]. ESCRT proteins are organized in functionally distinct complexes that drive an inverse membrane remodeling during various cellular processes, including cytokinetic abscission, vesicle biogenesis inside multivesicular endosomes, and viral budding, in addition to membrane repair[6–8]. All these processes require ESCRT-III proteins that form filaments within membrane invaginations and cooperate with the ATPase VPS4 to catalyze membrane constriction and fission away from the cytosol[9].

Activation of ESCRT enables cells to prevent potentially lethal consequences of minor perturbations in lysosomal integrity via a mechanism that seems to sense more subtle membrane injuries than galectins. For instance, a drop in membrane tension appears to promote ESCRT-III recruitment and intralumenal vesicle formation on endolysosomal membranes[10]. However, the recruitment signal that triggers ESCRT-III assembly at sites of lysosomal damage has not been established with certainty. While detection of $Ca^{2+}$ leakage out of the injured lysosome by ALIX and its $Ca^{2+}$-binding partner ALG2 has been proposed as a mechanism[4], other studies were unable to confirm a requirement for $Ca^{2+}$ and found that the ESCRT-I subunit TSG101 is more important than ALIX for mediating ESCRT-III recruitment to sites of drug- or pathogen-induced membrane injuries[5,11]. Thus, ESCRT assembly on damaged organelles may involve hitherto uncharacterized cues[6].

While glycans reside exclusively on the non-cytosolic surface of lysosomes and the plasma membrane, also certain lipids display strict asymmetric distributions across organellar bilayers. For instance, sphingomyelin (SM) is highly enriched in the exoplasmic leaflet of the plasma membrane whereas phosphatidylserine (PS) is primarily located in the cytosolic leaflet[12–14]. Translocation of PS to the outer leaflet during apoptosis marks dying cells and leads to their timely removal[15]. Application of a GFP-tagged version of lysenin, a SM-specific toxin from the earthworm *Eisenia fetida*[16], revealed that SM becomes exposed to the cytosol upon endomembrane damage caused by Gram-negative pathogens like *Shigella flexneri* or *Salmonella enterica*[17]. Breakout of these pathogens from the host vacuole into the cytosol follows a multi-step process, in which cytosolic SM exposure detected by the lysenin-based reporter invariably preceded glycan exposure, catastrophic membrane damage, and cytosolic entry of the bacteria. This raised the idea that the arrival of SM on the cytosolic surface of bacteria-containing vacuoles provides an early warning signal to alert cells of an imminent breakdown of organellar integrity[17]. How SM transfer across the bilayer of injured organelles is initiated and whether this process is part of a mechanism that helps preserve the integrity of cellular organelles remain to be explored.

In this work, we addressed the mechanism and functional implications of damage-induced SM scrambling. Using an engineered version of equinatoxin II as an SM reporter, we find that lysosome or plasma membrane wounding by chemicals, pathogens or light in each case triggers a rapid transbilayer movement of SM and that this process is $Ca^{2+}$-dependent, involving organelle-specific $Ca^{2+}$-activated scramblases. We show that SM removal promotes host endomembrane damage during *Salmonella* infection and impairs the recovery of lysosomes from acute injuries without affecting ESCRT recruitment. Moreover, we find that ectopic expression of a bacterial SMase targeted to the cytosolic surface of damaged lysosomes promotes their repair, also when ESCRT recruitment is blocked. Conversely, blocking endogenous neutral SMases disrupts lysosomal repair. We postulate that a $Ca^{2+}$-induced SM scrambling and hydrolysis drive an ESCRT-independent mechanism to clear minor lesions from the lysosome-limiting membrane and preserve lysosome function upon damage.

## Results

**SM is readily exposed to the cytosolic surface of damaged organelles.** To study how membrane damage triggers transbilayer movement of SM, we used an engineered version of the SM-binding pore-forming toxin, equinatoxin II (Eqt). Expression of EqtSM carrying a *N*-terminal signal sequence and *C*-terminal GFP tag previously enabled us to demonstrate sorting of native SM at the *trans*-Golgi network into a distinct class of secretory vesicles[18]. When expressed without signal sequence, EqtSM displayed a diffuse distribution throughout the cytosol and nucleus of HeLa cells (Fig. 1a, 0 min time point). Occasionally, the cytosolic reporter gave rise to a few small intracellular puncta, which may originate from a minor tendency of Eqt to aggregate upon overexpression or SM exposure following spontaneous membrane damage. However, in cells treated with L-leucyl-L-leucine O-methyl ester (LLOMe), a lysosomotropic compound commonly used to disrupt lysosome integrity[19], EqtSM underwent a massive accumulation in numerous puncta distributed throughout the cell within minutes after drug addition (Fig. 1a, b). These puncta displayed substantial overlap with LAMP1-positive compartments (Supplementary Fig. 1). LLOMe failed to trigger recruitment of EqtSol, a EqtSM variant defective in SM binding[18]. LLOMe-induced mobilization of EqtSM was virtually abolished upon genetic ablation of the SM synthases SMS1 and SMS2 (SMS-KO; Fig. 1a, b and Supplementary Fig. 2). While SMS removal essentially eliminated the cellular SM pool (Supplementary Fig. 2d), it did not impair the ability of LLOMe to disrupt lysosomal integrity (Supplementary Fig. 3). Collectively, these data indicate that EqtSM faithfully reports cytosolic exposure of SM by LLOMe-damaged lysosomes.

To analyze the kinetics of EqtSM mobilization in relation to the degree of lysosomal damage, we used cells co-expressing the SM reporter with mCherry-tagged Galectin-3 (Gal3), a cytosolic lectin with affinity for the complex glycans that reside on the non-cytosolic surface of lysosomes[20]. Upon LLOMe treatment, EqtSM readily accumulated in numerous puncta that also became gradually positive for Gal3 (Fig. 1c and Supplementary Movie 1). Importantly, EqtSM was recruited to LLOMe-damaged lysosomes prior to Gal3, with a time difference of ~5 min (Fig. 1d). As mCherry-Gal3 and EqtSM-GFP have similar molecular weights (53.8 and 49.3 kDa, respectively), it appears unlikely that this time gap is due to a size-dependent entry of the reporter molecules into the perforated organelles. Instead, our data are consistent with the study by Ellison et al.[17] and indicate that a break in SM asymmetry is an early sign of lysosomal damage that precedes a

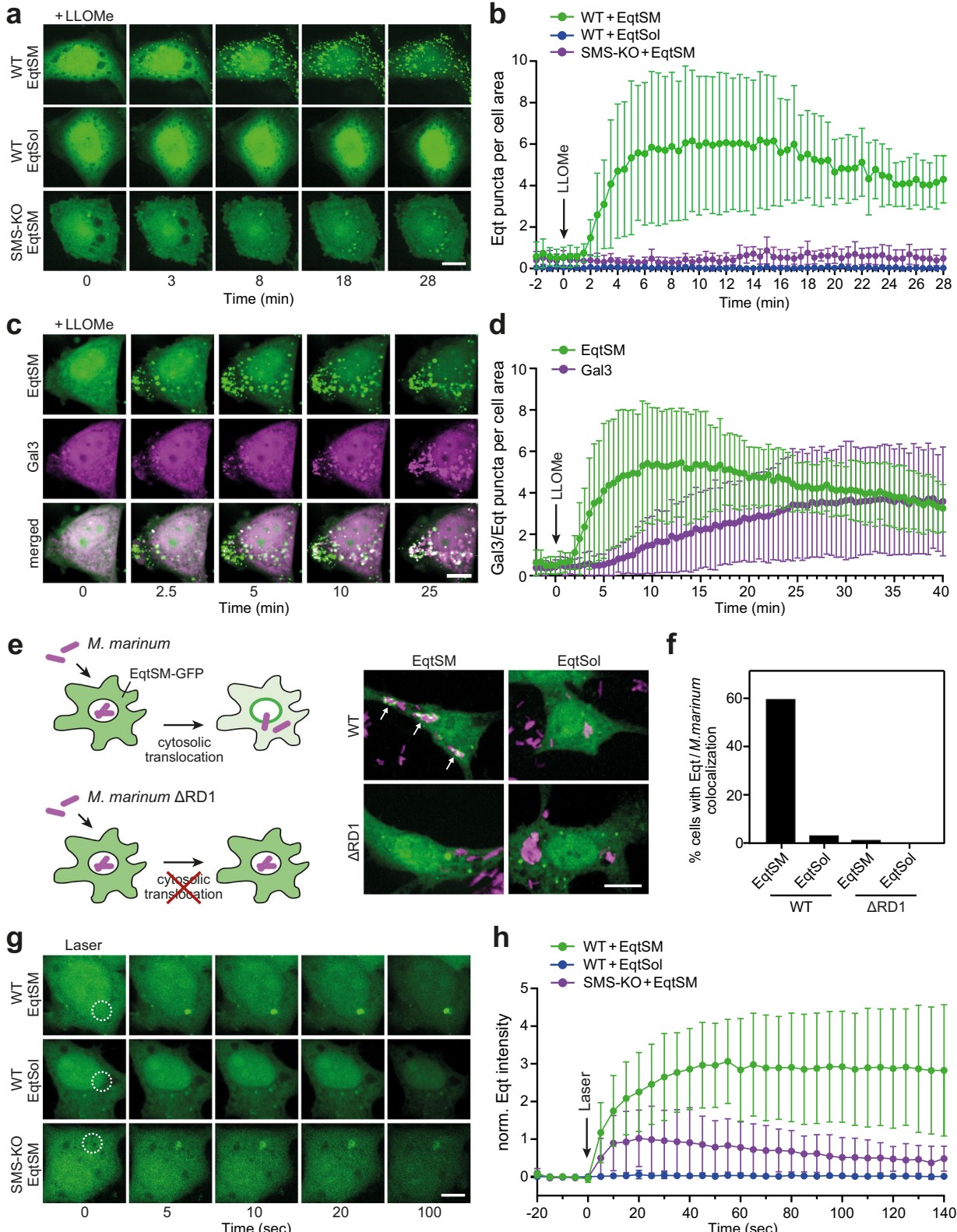

catastrophic breakdown of membrane integrity, when galectins gain access to the luminal glycans. Live-cell imaging of RAW246.7 macrophages invaded by the bacterial pathogen *Mycobacterium marinum* revealed strong recruitment of EqtSM to a subset of bacteria at 1–2 h post-infection (Fig. 1e, f and Supplementary Movie 2). No EqtSM recruitment was observed in macrophages invaded by the nonpathogenic *M. marinum* strain

ΔRD1, which fails to translocate to the cytosol and remains confined to the bacteria-containing phagosome due to a nonfunctional ESX-1 secretion system required for niche-breakage and pore-forming activity[21,22].

To further challenge a fundamental link between transbilayer SM movement and membrane damage, we next analyzed the distribution of EqtSM in cells exposed to distinct modes of plasma

**Fig. 1 Cytosolic EqtSM readily binds organelles injured by chemicals, pathogens, or light. a** Time-lapse images of wild-type (WT) or SMS-KO HeLa cells expressing GFP-tagged EqtSM or EqtSol and treated with 1 mM LLOMe for the indicated time. **b** Time-course plotting Eqt-positive puncta per 100 μm² cell area in cells treated as in (**a**). Data are means ± SD. $n = 9$ cells for WT + EqtSM, 9 cells for WT + EqtSol, 8 cells for SMS-KO + EqtSM over two independent experiments. **c** Time-lapse images of wild-type HeLa cells co-expressing GFP-tagged EqtSM (*green*) and mCherry-tagged Gal3 (*magenta*) treated with 1 mM LLOMe for the indicated time. **d** Time-course plotting Eqt- and Gal3-positive puncta per 100 μm² cell area in cells treated as in (**c**). Data are means ± SD. $n = 17$ cells. **e** RAW264.7 cells expressing GFP-tagged EqtSM or EqtSol (*green*) were infected with mCherry-expressing wild-type (WT) or translocation-defective (ΔRD1) mutant strains of *Mycobacteria marinum* (*magenta*). Live-cell fluorescence micrographs were captured 2 h post-infection. **f** Percentage of cells treated as in (**e**) showing co-localization of mCherry-expressing *M. marinum* and EqtSM-positive puncta. $n = 30$ cells per condition over two independent experiments. **g** Time-lapse images of wild-type (WT) or SMS-KO HeLa cells expressing GFP-tagged EqtSM or EqtSol and locally wounded by a brief pulse from a 2-photon laser. **h** Time-course plotting Eqt-associated fluorescence at the laser-induced wound site in cells treated as in (**g**). Data are means ± SD. $n = 7$ cells for WT + EqtSM, 4 cells for WT + EqtSol, 6 cells for SMS-KO + EqtSM over two independent experiments. Scale bar, 10 μm.

membrane wounding. To this end, we first incubated cells expressing EqtSM with the bilayer-destabilizing compound digitonin. We observed a redistribution of the cytosolic SM reporter to discrete puncta at the plasma membrane within minutes after transient exposure to the compound (Supplementary Fig. 4a). Similar results were obtained after prolonged exposure of cells to the pore-forming bacterial toxin streptolysin O (SLO; Supplementary Fig. 4b, c). To couple localized plasma membrane injuries to a fast imaging of downstream events, we next conducted laser-based plasma membrane wounding using a confocal scanning microscope equipped with a pulsed laser. This revealed an ultra-fast (within 5 s) and localized mobilization of EqtSM, but not EqtSol, to the laser-induced wound site (Fig. 1g, h and Supplementary Movie 3). While EqtSM was instantly recruited to the damaged membrane area in both wild-type and SMS-KO cells, the signal in the latter readily faded, in line with our finding that SMS-KO cells contain only residual amounts of SM (Supplementary Fig. 2d). Thus, a breach in membrane integrity caused by pore-forming chemicals, bacterial toxins or a laser appears to be tightly linked with a rapid transbilayer movement of SM.

**Damage triggers SM translocation through calcium-activated scramblases**. An influx of calcium from the extracellular environment or calcium-storing organelles has been identified as a key trigger for a rapid repair of membrane injuries[23,24]. As binding of EqtSM to SM-containing liposomes is independent of calcium (Supplementary Fig. 5), we asked whether calcium plays a role in the damage-induced translocation of SM. To address this, we analyzed the distribution of cytosolic EqtSM in HeLa cells exposed to SLO in calcium-free medium supplemented with the calcium chelator EGTA. Removal of extracellular calcium greatly impaired SLO-induced formation of EqtSM-positive puncta (Fig. 2a, b and Supplementary Fig. 4c). Moreover, treatment of cells with the calcium ionophore ionomycin triggered an accumulation of EqtSM in numerous puncta, but only when calcium was present in the medium (Fig. 2c, d and Supplementary Fig. 1a). The plasma membrane of mammalian cells harbors a calcium-activated phospholipid scramblase, TMEM16F, which catalyzes phosphatidylserine (PS) externalization in response to elevated intracellular calcium[25]. We therefore wondered whether TMEM16F plays a role in SM scrambling at sites of plasma membrane damage. Strikingly, genetic ablation of TMEM16F abolished the calcium-dependent formation of EqtSM-positive puncta in both SLO- and ionomycin-treated cells (Fig. 2a–d and Supplementary Fig. 6). This indicates that TMEM16F is responsible for damage-induced SM movement across the plasma membrane.

Because lysosomes store calcium and rupturing them increases cytosolic calcium, we next asked whether EqtSM recruitment to damaged lysosomes is similarly controlled by calcium-activated scramblases. Preloading cells with the cell-permeant calcium chelator BAPTA-AM significantly impaired recruitment of the

cytosolic SM reporter to LLOMe-injured lysosomes (Fig. 2e, f). However, removal of TMEM16F did not affect the lysosomal damage-induced mobilization of the reporter (Fig. 2e, g). Based on these results, we conclude that the transbilayer movement of SM in response to membrane injuries is a calcium-dependent process, involving TMEM16F at the plasma membrane and presumably a related scramblase in lysosomes (Fig. 2h). The identity of the lysosomal scramblase remains to be established.

**SM-deficient cells are defective in lysosomal repair**. To investigate whether SM participates in the repair of damaged lysosomes, we took advantage of the ability of lysosomes to retain LysoTracker, a weak base that accumulates in the organelle's acidic lumen and is fluorescent at low pH[26]. LysoTracker fluorescence is lost upon brief exposure of cells to the lysosomotropic compound glycyl-L-phenylalanine 2-naphtylamide (GPN) but returns after GPN is washed away (Fig. 3a, b)[4]. Like LLOMe, GPN induces a transient disruption of lysosomal integrity that is accompanied by an accumulation of EqtSM in numerous puncta (Fig. 3b, c and Supplementary Movie 4). However, GPN is processed into metabolites thought to promote osmotic rupture[27]. As lysosomal pH critically relies on the structural integrity of the membrane, the loss and recovery of LysoTracker fluorescence in cells transiently exposed to GPN is consistent with lysosomal membrane disruption and restoration. Wild-type and SMS-KO cells acquired LysoTracker to a similar extent (Supplementary Fig. 7a, b), and also lost LysoTracker fluorescence with similar kinetics after the addition of GPN (Fig. 3d, e and Supplementary Fig. 7c), indicating comparable lysosome function. However, recovery of LysoTracker fluorescence after GPN removal was significantly delayed in the SMS-KO cells. Moreover, less LysoTracker-positive structures recovered in SMS-KO cells than in wild-type (Fig. 3d, e). In SMS-KO cells transduced with SMS1 under control of a doxycycline-inducible promotor, the addition of doxycycline restored the capacity to produce SM (Supplementary Fig. 8) and regain LysoTracker fluorescence to the same level as in wild-type cells after GPN is washed away (Fig. 3f, g). Likewise, the addition of SM-containing liposomes to SMS-KO cells not only restored EqtSM recruitment to LLOMe-damaged lysosomes (Fig. 3h) but also rescued the defect in lysosomal repair (Fig. 3i and Supplementary Fig. 7d). No such rescuing effects were observed in SMS-KO cells supplied with SM-deficient liposomes. SMS removal also caused a significant rise in cell death in the presence of LLOMe (Fig. 3j). Collectively, these data indicate that SM plays a critical role in the recovery of lysosomes from acute, potentially lethal damage.

**SM removal promotes host endomembrane damage during *Salmonella* infection**. Previous work revealed that pathogen-containing vacuoles are prone to sporadic ruptures, which are repeatedly repaired by the ESCRT machinery[5,11,28]. As SM

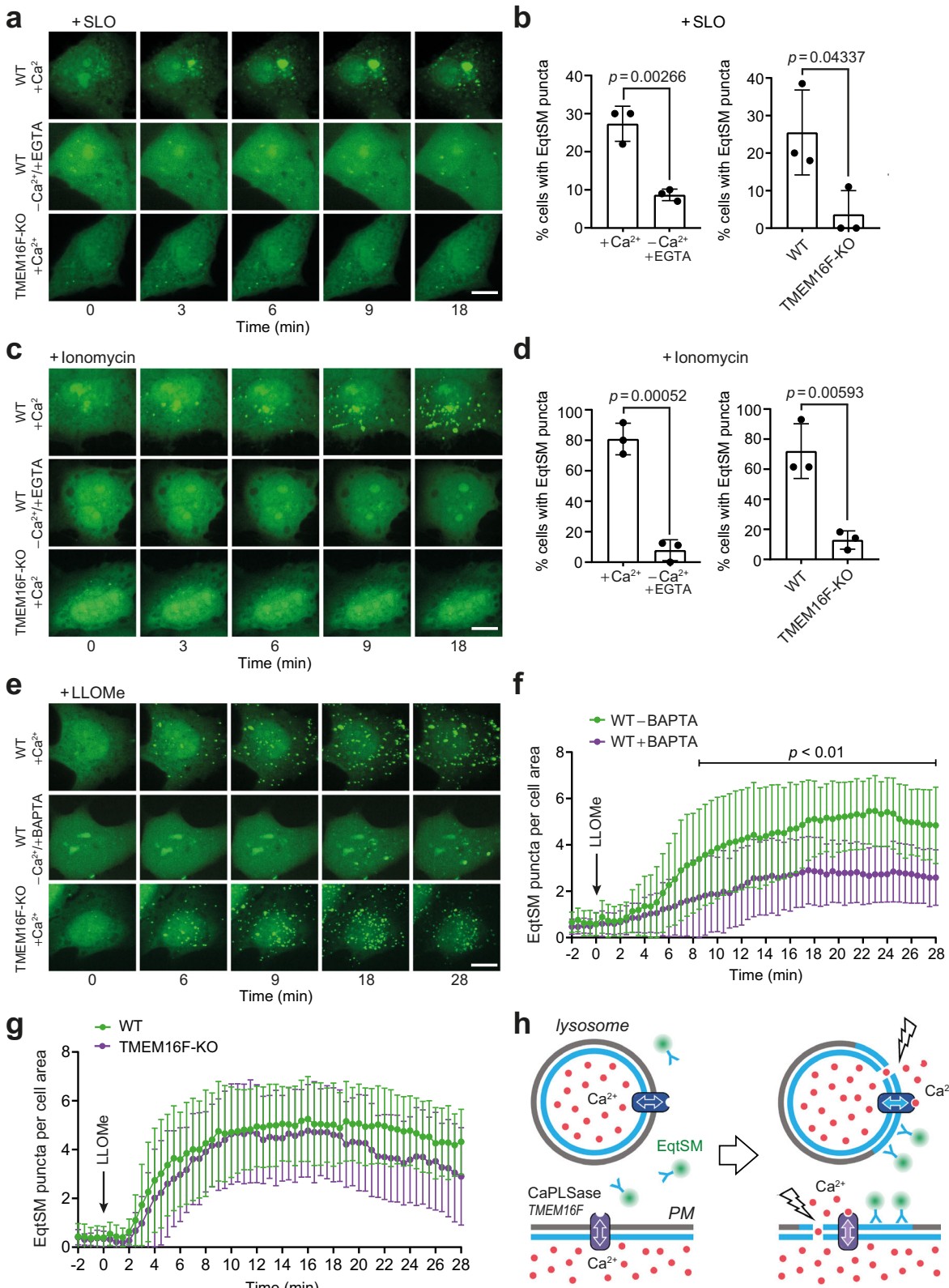

becomes exposed to the cytosol upon endomembrane damage caused by pathogens like *Salmonella enterica*[17] and *Mycobacterium marinum* (Fig. 1e), we wondered whether SM also participates in sustaining host vacuole integrity during bacterial infection. To address this, we infected HeLa cells expressing Halo-tagged EqtSM with *Salmonella* containing a dual-color reporter plasmid for constitutive expression of dsRed and encoding sfGFP

under control of the glucose-6-phosphate (G6P)-inducible promotor $P_{uhpT}$ (Fig. 4a). Based on the short maturation time of sfGFP (<14 min), the reporter enables a dynamic and sensitive detection of pathogen exposure to the host cytosol[29]. Using time-lapse microscopy of infected cells, we observed that mobilization of EqtSM to bacteria-containing vacuoles consistently preceded a collapse of vacuole integrity and expression of sfGFP (Fig. 4b and

**Fig. 2 Damage-induced SM translocation is mediated by a calcium-activated lipid scramblase. a** Time-lapse images of wild-type (WT) or TMEM16F-KO HeLa cells expressing GFP-tagged EqtSM and treated with 1500 U/ml SLO for the indicated time in medium containing ($+Ca^{2+}$) or lacking $Ca^{2+}$ ($-Ca^{2+}$/$+EGTA$). **b** Percentage of cells displaying EqtSM-positive puncta after 30 min of SLO treatment as in (**a**). Data are means ± SD. $n = 29$ cells for $+Ca^{2+}$, 35 cells for $-Ca^{2+}$/$+EGTA$, 35 cells for WT, 32 cells for TMEM16F-KO over three independent experiments. $P$ values were calculated by unpaired two-tailed $t$ test. **c** Time-lapse images of WT or TMEM16F-KO HeLa cells expressing GFP-tagged EqtSM and treated with 5 μM ionomycin for the indicated time in medium containing ($+Ca^{2+}$) or lacking $Ca^{2+}$ ($-Ca^{2+}$/$+EGTA$). **d** Percentage of cells displaying EqtSM-positive puncta after 30 min of ionomycin treatment as in (**a**). Data are means ± SD. $n = 41$ cells for $+Ca^{2+}$, 25 cells for $-Ca^{2+}$/$+EGTA$, 41 cells for WT, 41 cells for TMEM16F-KO over three independent experiments. $P$ values were calculated by unpaired two-tailed $t$ test. **e** Time-lapse images of WT or TMEM16F-KO HeLa cells expressing GFP-tagged EqtSM and treated with 1 mM LLOMe after preincubation with or without BAPTA-AM (100 μM, 45 min). **f** Time-course plotting EqtSM-positive puncta per 100 μm² cell area in wild-type (WT) cells treated as in (**e**). Data are means ± SD. $n = 24$ cells for WT −BAPTA, 21 cells for WT + BAPTA over three independent experiments. $P$ values were calculated by unpaired two-tailed $t$ test. **g** Time-course plotting EqtSM-positive puncta per 100 μm² cell area in wild-type or TMEM16F-KO HeLa cells treated with LLOMe in the absence of BAPTA-AM as in (**f**). Data are means ± SD. $n = 25$ cells for WT, 15 cells for TMEM16F-KO over three independent experiments. **h** Schematic illustration of how membrane damage triggers SM scrambling. PM plasma membrane, CaPLSase calcium-activated phospholipid scramblase. Scale bar, 10 μm. Source data including exact $P$ values are provided as a Source Data file.

Supplementary Movie 5), hence in line with the study by Ellison et al.[17] and the data presented in Fig. 1d. Combining the dual-color reporter approach with quantitative flow cytometry, we found that SM removal enhances vacuole disruption and cytosolic release of *Salmonella* during host cell infection to a similar degree as loss of CHMP3, an essential subunit of the ESCRT-III complex (Fig. 4c, d). These results reinforce the notion that SM is a core component of a pathway used by cells to preserve the integrity of their endomembranes in the face of a wide variety of insults.

**SM is dispensable for ESCRT recruitment to damaged lysosomes.** As ESCRT recruitment is at least partially dependent on a rise in intracellular calcium[4], we wondered whether cytosolic SM exposure may be part of the mechanism by which ESCRT is mobilized to injured lysosomes. To address this idea, we first monitored the subcellular distribution of both EqtSM and CHMP4B, an ESCRT-III component necessary for all known mammalian ESCRT functions[6], in LLOMe-treated cells using time-lapse microscopy. This revealed that mobilization of EqtSM and CHMP4B to LLOMe-damaged lysosomes occurred with very similar kinetics (Fig. 5a, b and Supplementary Movie 6). We then asked whether SM is essential for CHMP4B recruitment. However, the rate and efficiency by which CHMP4B accumulated on LLOMe-damaged lysosomes in SMS-KO cells were indistinguishable from those in wild-type cells (Fig. 5c, d). On the other hand, SMS-KO cells displayed a prolonged CHMP4B retention on damaged lysosomes, a finding consistent with a critical role of SM in lysosomal repair. Moreover, siRNA-mediated depletion of ALIX and TSG101, two proteins essential for ESCRT-III recruitment to damaged lysosomes[5], further reduced the viability of LLOMe-treated SMS-KO cells while having only a minor impact on the survival of LLOMe-treated controls (Fig. 3i, j and Supplementary Fig. 9). This indicates that cells are equipped with an SM-dependent lysosomal repair pathway that operates in parallel with the ESCRT pathway.

**Hydrolysis of cytosolic SM enhances lysosomal repair in ESCRT-compromised cells.** To investigate whether SM exposure on the cytosolic surface of damaged lysosomes is critical for their repair, we generated a construct in which a SMase from *Bacillus cereus* was fused to the cytosolic tail of the lysosome-associated membrane protein LAMP1 (Fig. 6a). The LAMP1-bSMase fusion protein co-localized with both LAMP1-positive and LysoTracker-labeled compartments and catalyzed the metabolic conversion of fluorescent NBD-SM into NBD-ceramide (Supplementary Fig. 10). LAMP1 fused to a catalytically inactive bSMase mutant (D322A/H323A) served as control. Live-cell imaging revealed that expression of active LAMP1-bSMase, but not its enzyme-

dead counterpart, efficiently suppressed mobilization of cytosolic EqtSM to LLOMe-damaged lysosomes (Fig. 6b, c and Supplementary Movies 7 and 8), indicating that the active fusion protein catalyzes a rapid and efficient hydrolysis of SM on the cytosolic surface of damaged lysosomes. To address whether metabolic turnover of cytosolic SM reduces or enhances the repair of damaged lysosomes, we next analyzed the impact of active or enzyme-dead LAMP1-bSMase on the ability of cells to regain LysoTracker fluorescence after transient exposure to GPN. We found that expression of active LAMP1-bSMase enhanced rather than diminished the recovery of LysoTracker fluorescence in GPN-treated cells (Fig. 6d). This effect became even more prominent when these experiments were carried out on cells treated with ALIX and TSG101-targeting siRNAs (Fig. 6e). Thus, a calcium-induced exposure and subsequent metabolic turnover of SM on the cytosolic surface of damaged lysosomes appears to promote their repair, even when ESCRT recruitment is blocked.

**Inhibition of neutral SMases disrupts lysosomal repair.** As ectopic expression of a bacterial SMase targeted to the cytosolic surface of lysosomes enhanced lysosomal repair, we anticipated that inhibition of endogenous neutral SMases, which act on cytosolic SM pools, might perturb lysosomal repair. Indeed, the addition of the generic neutral SMase inhibitor GW4869 significantly impaired the recovery of LysoTracker fluorescence in cells transiently exposed to GPN (Fig. 7a). The negative impact of GW4869 on lysosomal repair was even more pronounced in cells in which ESCRT function was compromised by pre-treatment with ALIX and TSG101-targeting siRNAs (Fig. 7b). Consistent with a critical role of neutral SMases in lysosomal repair, GW4869 significantly reduced the viability of both control and ALIX/TSG101-depleted cells in the presence of LLOMe (Fig. 7c, d). This raised the question which of the known neutral SMase isoforms participates in the restoration of damaged lysosomes. Four mammalian neutral SMases have been identified to date, namely nSMase-1 (SMPD2), nSMase-2 (SMPD3), nSMase-3 (SMPD4), and mitochondria-associated MA-nSMase (SMPD5)[30]. While the expression of nSMase-3 is mainly restricted to skeletal muscle and heart[31], nSMase-1 and -2 are ubiquitously expressed and promote SM hydrolysis on the cytosolic surface of the ER and plasma membrane, respectively[32,33]. Using the LysoTracker fluorescence recovery assay, we found that siRNA-mediated depletion of nSMase-1 had no impact on the repair of GPN-damaged lysosomes in ESCRT-compromised cells (Fig. 7e). In contrast, siRNA-mediated depletion of nSMase-2 significantly impaired the recovery of lysosomes injured by GPN (Fig. 7f and Supplementary Fig. 11a). Moreover, genetic ablation of nSMase-2 recapitulated both the defect in lysosomal repair and enhanced sensitivity toward LLOMe observed in GW4869-treated

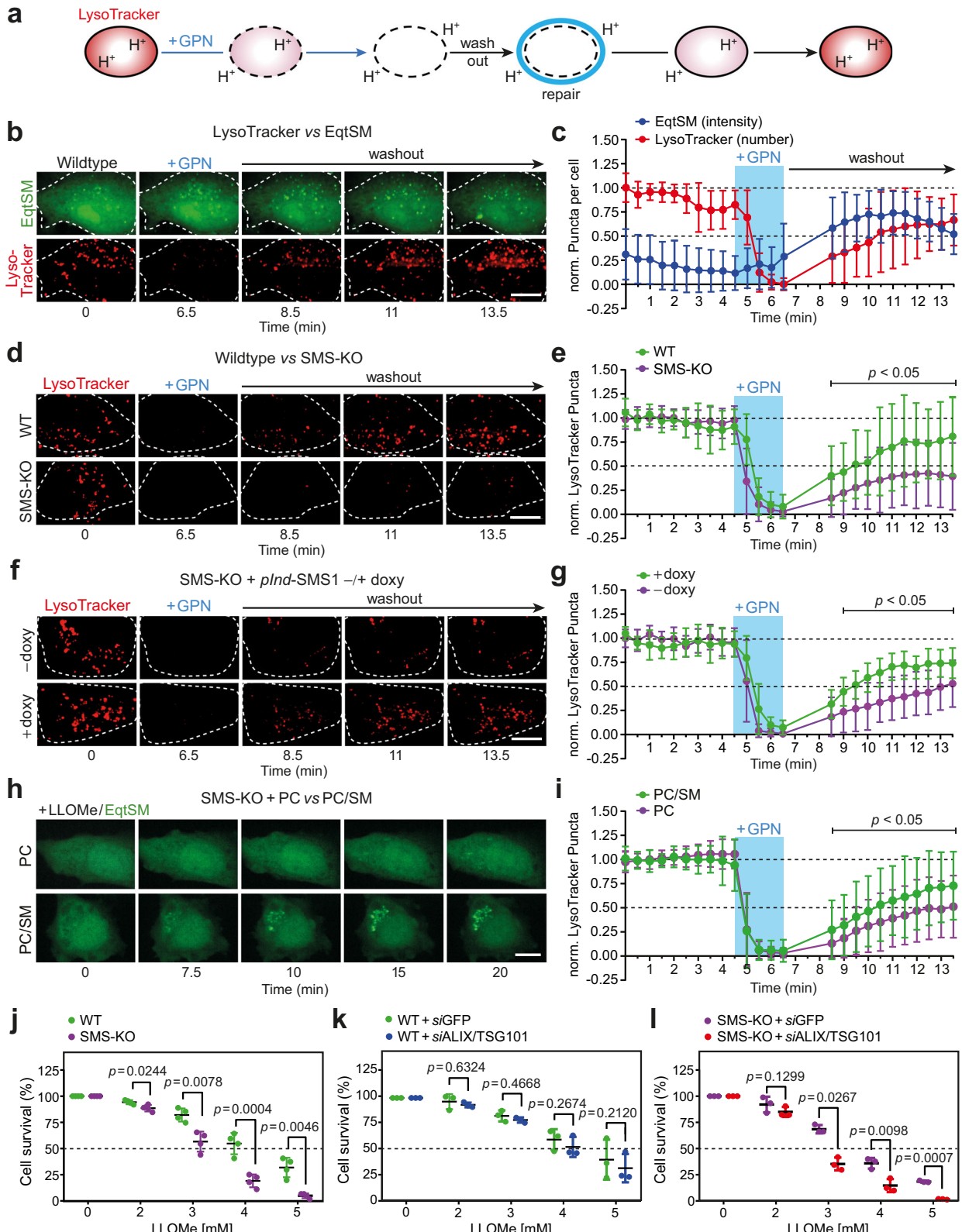

cells (Fig. 7g, h). This indicates that nSMase-2 is a critical component of an SM-dependent membrane repair pathway for the restoration of damaged lysosomes.

## Discussion

The ESCRT machinery plays a well-established role in responding to and repairing damaged lysosomes. Here, we uncovered a complementary sphingolipid-operated lysosomal repair pathway that reverses potentially lethal membrane damage inflicted by lysosomotropic peptides and restores compartmental pH independently of ESCRT. The main features of this membrane repair pathway are depicted in Fig. 7g. We envision that minor perturbations in the integrity of the lysosomal membrane cause calcium ions to leak from the organelle's lumen into the cytosol.

**Fig. 3 SM is critical for the recovery of lysosomes from acute damage. a** Schematic outline of lysosomal repair assay. **b** Time-lapse images of LysoTracker-labeled/EqtSM-expressing HeLa cells pulse-treated with GPN (200 μM, 2 min). **c** Time-course plotting LysoTracker- and EqtSM-positive puncta in cells treated as in (**b**). Data are means ± SD. $n = 18$ cells over four independent experiments. **d** Time-lapse images of LysoTracker-labeled WT and SMS-KO HeLa cells pulse-treated with GPN. **e** Time-course plotting LysoTracker-positive puncta in cells treated as in (**b**). Data are means ± SD. $n = 29$ cells for WT, 24 cells for SMS-KO over four independent experiments. **f** Time-lapse images of LysoTracker-labeled SMS-KO cells transduced with doxycycline-inducible SMS1 (pInd-SMS1) and pulse-treated with GPN following 48 h preincubation without (–doxy) or with 1 μg/ml doxycycline (+doxy). **g** Time-course plotting LysoTracker-positive puncta in cells treated as in (**d**). Data are means ± SD. $n = 20$ cells for +doxy, 17 cells for –doxy over three independent experiments. **h** Time-lapse images of SMS-KO cells expressing EqtSM treated with 1 mM LLOMe for the indicated time following 16 h preincubation with 160 μM PC or PC/SM (70/30 mol%). **i** Time-course plotting LysoTracker-positive puncta in SMS-KO cells pulse-treated with GPN following preincubation with PC or PC/SM as in (**h**). Data are means ± SD. $n = 50$ cells per condition over three independent experiments. **j** Survival rate of WT and SMS-KO cells after 5 h exposure to LLOMe at the indicated concentration. Data are means ± SD. $n = 4$ independent experiments. **k** Survival rate of WT cells pre-treated with siRNAs targeting GFP or ALIX/TSG101 for 72 h and exposed for 5 h to LLOMe at indicated concentrations. Data are means ± SD. $n = 3$ independent experiments. **l** Survival rate of SMS-KO cells pre-treated with siRNAs targeting GFP or ALIX/TSG101 for 72 h and exposed for 5 h to LLOMe at indicated concentrations. Data are means ± SD. $n = 3$ independent experiments. $P$ values were calculated by unpaired (**e**, **g**, **i**) or paired two-tailed $t$ test (**j**, **k**, **l**). Scale bar, 10 μm.

A local rise in cytosolic calcium triggers calcium-activated scramblases near the injury site, resulting in a rapid exposure of SM on the cytosolic surface of the damaged organelle. We propose that the expanding pool of SM in the cytosolic leaflet is turned over by a neutral SMase, presumably nSMase-2, even though we do not rule out an involvement of other members of the nSMase enzyme family. Neutral SMases cleave the bulky phosphorylcholine head group of SM to generate ceramide, a lipid with a cone-shaped structure that occupies a smaller membrane area than SM. Ceramides released by SM turnover readily self-assemble into microdomains that possess a negative spontaneous curvature[34]. Consequently, nSMase-mediated conversion of SM to ceramide would cause a local condensation of the cytosolic leaflet near the site of membrane injury. This, in turn, would promote an inverse budding of the bilayer away from the cytosol, akin to the ESCRT-mediated formation of intraluminal vesicles[35]. Collectively, our data suggest that a calcium-induced SM scrambling and turnover drives an ESCRT-independent mechanism to clear minor lesions from the lysosome-limiting membrane and prevent lysosomal damage-induced cell death. In line with our findings, ceramide-based membrane invaginations were demonstrated in SM-containing giant liposomes exposed to external SMases[36,37] and have been implicated in the biogenesis of proteolipid-containing exosomes inside multivesicular endosomes, a process occurring independently of ESCRT and with nSMase-2 playing a crucial role[38,39].

Lysosomal acid SMase (aSMase) previously emerged as a key player in the repair of plasma membrane damage caused by pore-forming toxins. Thus, aSMase has been shown to promote plasma membrane invagination and endocytosis in SLO-permeabilized cells in response to $Ca^{2+}$-triggered exocytosis of lysosomes[40]. Here, removal of SLO-damaged plasma membrane areas relies on ceramide microdomain formation in the exoplasmic leaflet through aSMase-mediated hydrolysis of SM, which is normally concentrated there. While this process is mediated by a classical budding of the bilayer toward the cytosol, there is also evidence for alternative plasma membrane repair pathways in which lesions are removed by a reverse budding and shedding of extracellular vesicles. For instance, real-time imaging and correlative scanning electron microscopy of cells wounded by a laser provided evidence for ESCRT-mediated extracellular shedding of the damaged plasma membrane area[7]. Moreover, recent work revealed that TMEM16F promotes plasma membrane repair in cells exposed to pore-forming toxins by facilitating the release of extracellular vesicles to eliminate the toxin from the membrane[41]. This raised the idea that PS exposure catalyzed by TMEM16F helps protect cells from external attacks and injuries by constituting a "repair me" signal. However, our present findings raise

an alternative scenario in which an injury-induced scrambling of SM mediated by TMEM16F serves to fuel a sphingolipid-based membrane restoration pathway analogous to the one that operates in lysosomes. Thus, the striking SM asymmetry that marks late secretory and endolysosomal organelles may actually reflect a vital role of SM and neutral SMases in safeguarding the functional integrity of these organelles.

## Methods

**Chemical reagents**. Chemical reagents were used at the following concentrations, unless indicated otherwise: 1 mM L-leucyl-L-leucine O-methyl ester (LLOMe; Bachem, 4000725); 200 μM glycyl-L-phenylalanine 2-naphtylamide (GPN; Abcam, ab145914); 1500 U/ml streptolysin O (SLO; Sigma-Aldrich, S5265); 250 μM digitonin (Sigma-Aldrich, D141); 5 μM ionomycin (Sigma-Aldrich, I0634); 100 μM BAPTA-AM (Cayman Chemical, 15551); 75 nM LysoTracker™ Red DND-99 (Thermo Fisher Scientific; L7528); 1 μg/ml doxycycline (Sigma-Aldrich;); and 10 μM GW4869 (Sigma-Aldrich, D1692). GW4869 was stored at −80 °C as a 2 mM stock suspension in dimethyl sulfoxide (DMSO). Just before use, the suspension was solubilized by the addition of 5% methane sulfonic acid as described previously ref. [42].

**Antibodies**. Antibodies used were: rabbit polyclonal anti-TMEM16F (Sigma-Aldrich, HPA038958; IB 1:1000); mouse monoclonal anti-SMS2 [7D10] (Santa Cruz, sc-293384; IB 1:1000); mouse monoclonal anti-nSMase-2 (Santa Cruz, sc-166637; IB: 1:1000); mouse monoclonal anti-LAMP1 [H4A3] (Santa Cruz, sc-20011; IF 1:200); rabbit polyclonal anti-CHMP4B (Proteintech, 13683-1-AP; IF 1:300); mouse monoclonal anti-ALIX [3A9] (Biolegend, 634501, IB 1:1000); mouse monoclonal anti-Actin (Sigma-Aldrich, A1978; IF 1:1200; IB 1:10,000); rabbit monoclonal anti-Na/K-ATPase [EP1845Y] (Abcam, ab-76020; IF 1:600); mouse monoclonal anti-TSG101 [C-2] (Santa Cruz, sc-7964; IB 1:1000); mouse monoclonal anti-V5 [R960-25] (Invitrogen, r96025; IF 1:400; IB 1:1000); rabbit polyclonal anti-GFP (Novus Biologicals, NB600-303; IF 1:250); HRP-conjugated goat anti-mouse IgG (Thermo Fisher Scientific, 31430; IB 1:5000); HRP-conjugated goat anti-rabbit IgG (Thermo Fisher Scientific; 31460; IB 1:5000); Cyanine Cy™2-conjugated donkey anti-mouse IgG (Jackson ImmunoResearch Laboratories, 715-225-150; IF 1:400); Cyanine Cy™2-conjugated donkey anti-rabbit IgG (Jackson ImmunoResearch Laboratories, 711-225-152; IF 1:400); Cyanine Cy™3-conjugated donkey anti-rabbit IgG (Jackson ImmunoResearch Laboratories, 715-165-152; IF 1:400); and Cyanine Cy™3-conjugated donkey anti-mouse IgG (Jackson ImmunoResearch Laboratories, 715-165-150; IF 1:400).

**DNA constructs**. Expression constructs encoding cytoplasmic GFP-tagged EqtSM and EqtSol were created by PCR amplification of DNA encoding residues 22-227 of EqtSM or EqtSol[18] followed by cloning into EcoRI and BamHI sites of pN1-oxGFP[43]. Expression constructs encoding cytoplasmic mKate-tagged EqtSM and EqtSol were created by PCR amplification followed by cloning into NheI and AgeI sites of mKate-LifeAct-7 (Addgene, 54697), thereby replacing the LifeAct ORF. An expression construct encoding cytoplasmic EqtSM fused to a HaloTag was created by PCR amplification of DNA encoding HaloTag residues 3–297 followed by cloning into BamHI and NotI sites of pN1-EqtSM-oxGFP, thereby replacing the oxGFP ORF. Expression constructs encoding mCherry-tagged human Galactin3 (pLX304-mCherry-hGalectin3), GFP-tagged human LAMP1 (pCMV-hLAMP1-GFP) and mCherry-tagged human LAMP1 (pCMV-hLAMP1-mCherry) have been described previously[28,29]. Expression constructs encoding bacterial SMase fused to GFP-tagged LAMP1 were created by PCR amplification of DNA encoding residues 28-333 of *Bacillus cereus* SMase using pEF6-bSMase-V5-His and pEF6-bSMase$^{D322A/H323A}$-V5-His[44] as templates, followed by cloning into BamHI and AgeI sites of pCMV-

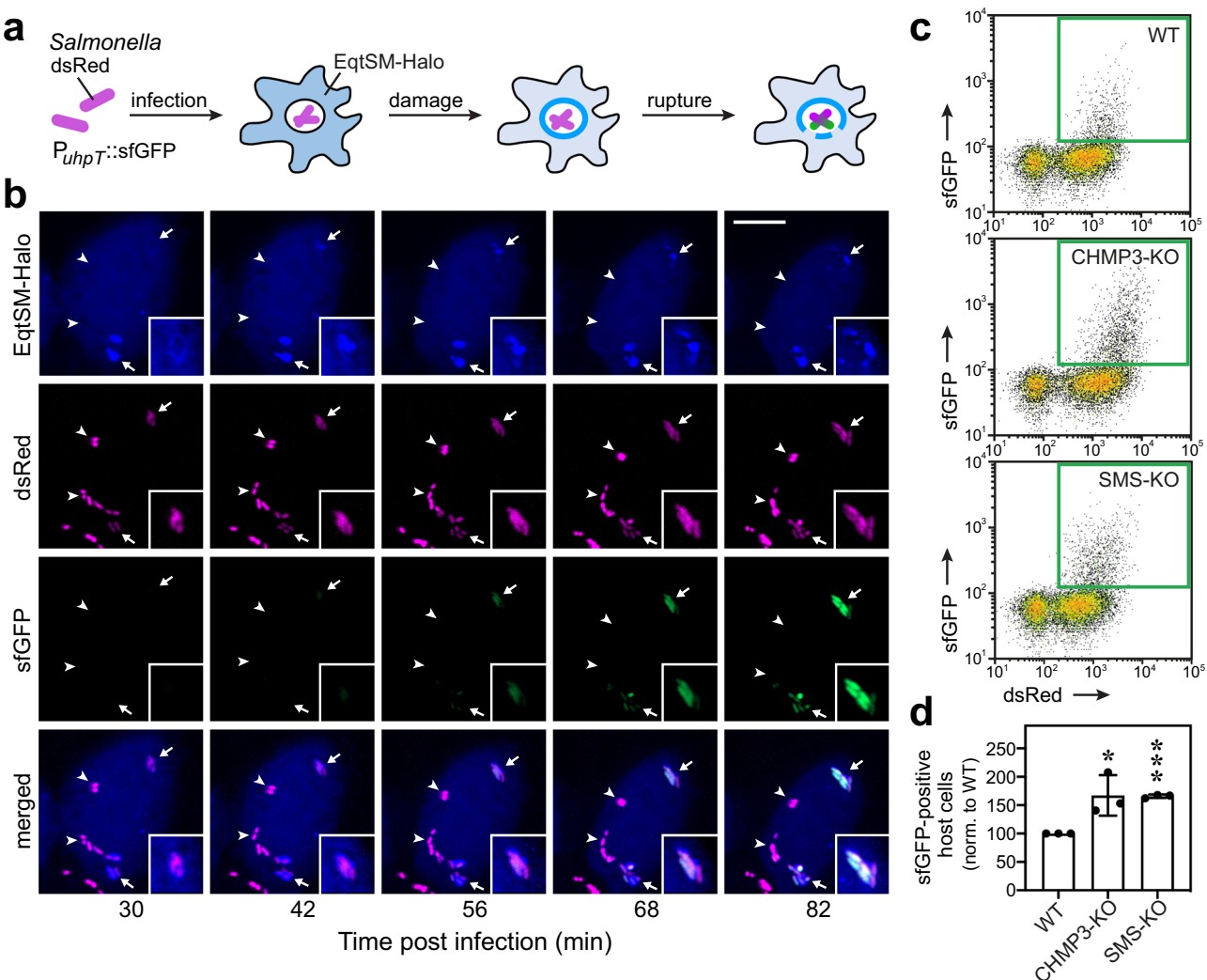

**Fig. 4 SM removal enhances vacuolar damage and cytosolic release of *Salmonella* during host cell infection. a** To quantify pathogen-induced vacuolar damage in a time-resolved manner, HeLa cells expressing Halo-tagged EqtSM were infected with *Salmonella* carrying a plasmid for constitutive expression of dsRed and encoding sfGFP under control of the glucose-6-phosphate (G6P)-inducible promotor $P_{uhpT}$. **b** Time-lapse images of HeLa cells expressing EqtSM-Halo labeled with silicone rhodamine (*blue*) and infected with *Salmonella* (dsRed, *magenta*) carrying the dual-fluorescence G6P-sensor plasmid as in (**a**). Note that *Salmonella* in EqtSM-positive vacuoles (*arrows*) eventually acquire GFP fluorescence (sfGFP, *green*), indicative of a collapse of vacuole integrity and exposure to host cytosol. In contrast, *Salmonella* in vacuoles devoid of EqtSM do not acquire GFP fluorescence (*arrowheads*). Scale bar, 10 μm. **c** Wild-type (WT), CHMP3-KO and SMS-KO HeLa cells infected with *Salmonella* carrying the dual-fluorescence G6P-sensor plasmid were subjected to flow cytometry at 3 h.p.i. using the gating strategy provided in Supplementary Fig. 12. The green square marks dsRed- and sfGFP-positive cells, which correspond to cells containing *Salmonella* exposed to host cytosol. **d** Cells infected with *Salmonella* were analyzed by flow cytometry at 3 h.p.i. to determine the proportion of infected cells (dsRed-positive) containing host cytosol-exposed bacteria (sfGFP-positive). All values were normalized to wildtype (WT). Data are means ± SD. $n = 3$ independent experiments performed in technical triplicates. $^*P = 0.03115$, $^{***}P = 3.04 \times 10^{-6}$ by unpaired two-tailed $t$ test.

hLAMP1-GFP to yield LAMP1-bSMase-V5-GFP and LAMP1-bSMase^dead^-V5-GFP. Expression construct pEF6-nSMase-2-V5 encoding V5-tagged human neutral SMase2 has been described previously[45]. A doxycycline-inducible expression construct encoding human SMS1 with a *N*-terminal FLAG tag (MDYKDDDDK) was created by PCR amplification using pcDNA1.3-SMS1-V5[46] as a template, followed by cloning into BamHI and NotI sites of pENTR™11 (Invitrogen, A10467). The insert was next transferred into lentiviral expression vector *pInducer20* (Addgene, 44012) using Gateway cloning, according to the manufacturer's instructions. The *Salmonella* dual-fluorescence reporter p4889 harboring $P_{EM7}$::dsRed for constitutive expression of dsRed and $P_{uhpT}$::sfGFP for sfGFP expression in the presence of glucose-6-phosphate has been described previously[29].

**Liposome binding assay.** Production of recombinant EqtSM and liposome binding assays were carried out as described previously[18]. In brief, EqtSM was expressed in *E. coli* BL21 (DE3) cells with an *N*-terminal polyHis tag from pET28a vector. Upon induction with 0.4 mM isopropyl β-ᴅ-1-thio-galactopyranoside (4 h, 37 °C), cells were harvested and mechanically lysed in 20 mM Na₂HPO₄/NaH₂PO₄, pH 7.4, 500 mM NaCl, 25 mM imidazole, and cOmplete™ EDTA-free Protease

Inhibitor Cocktail (Roche). Cleared lysates were applied to a HisTrap column using an AKTA Prime liquid chromatography system (GE Healthcare), and bound EqtSM was eluted with an imidazole gradient. Liposomes were produced from synthetic lipid mixtures containing 60 mol% dioleoyl-phosphatidylcholine (DOPC; Avanti Polar Lipids, 850375), 20 mol% cholesterol (Avanti Polar Lipids, 7000000) and 20 mol% porcine brain SM (Avanti Polar Lipids, 860062), or 80 mol% DOPC and 20 mol% cholesterol in liposome binding buffer (10 mM HEPES, 100 mM NaCl, pH 6.5) by extrusion at 65 °C through 1-μm-pore filters using a mini-extruder (Avanti Polar Lipids). For binding assays, liposomes (1 mM lipid) were supplemented with the indicated concentration of CaCl₂ and then incubated with purified EqtSM (10 μM) for 5 min at 37 °C. Liposomes were collected by centrifugation at $100,000 \times g$ for 10 min at 37 °C. Collected supernatants and pellets resuspended in an equal volume of liposome binding buffer were analyzed by SDS-PAGE and Coomassie brilliant blue staining.

**Cell culture and siRNA treatment.** Human cervical carcinoma HeLa (ATCC CCL-2) and human embryonic kidney HEK293T cells (ATCC CRL-3216) were cultured in Dulbecco's modified Eagle's medium (DMEM) supplemented with

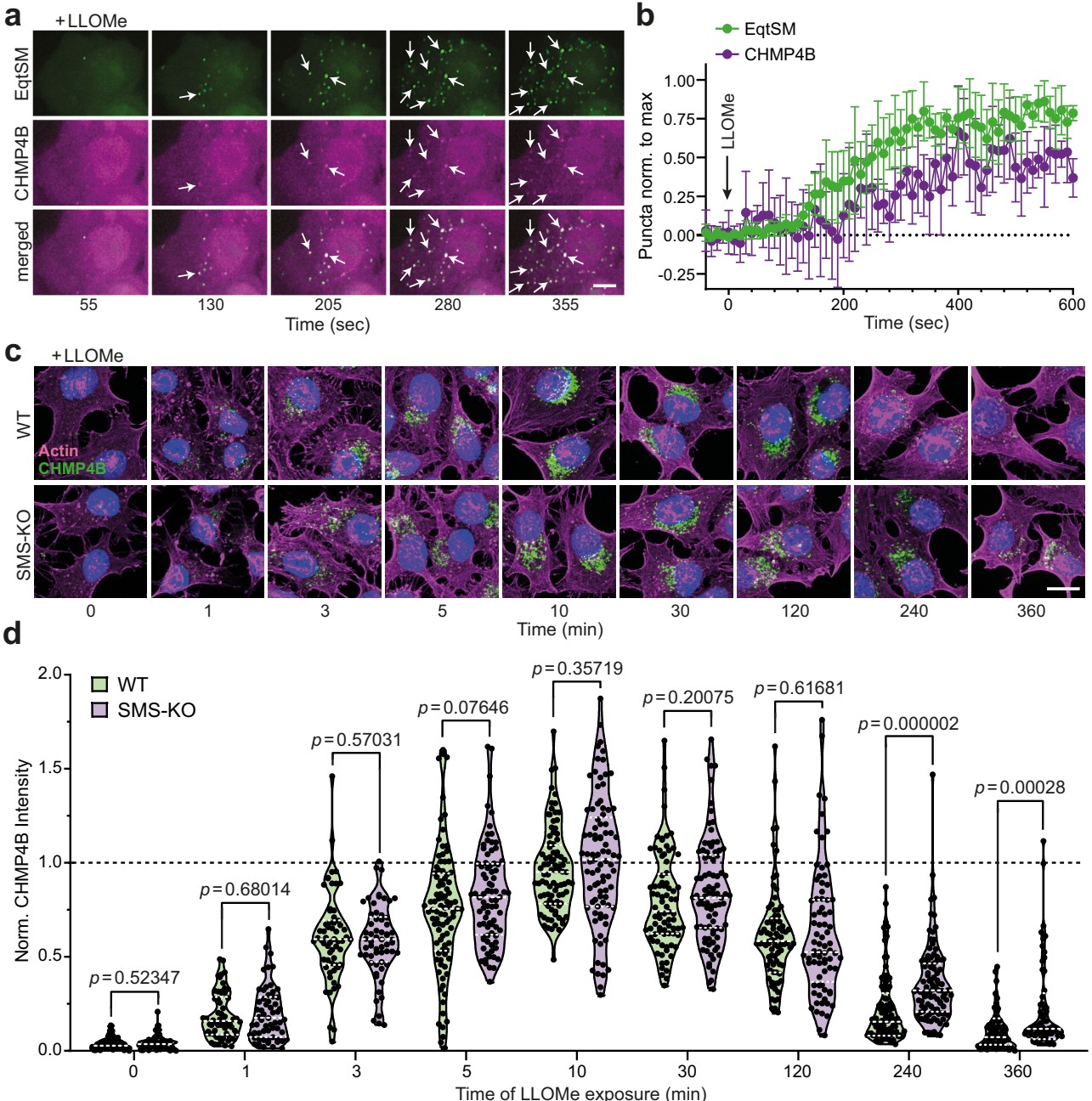

**Fig. 5 SM is dispensable for ESCRT recruitment to damaged lysosomes. a** Time-lapse images of HeLa cells co-expressing mKate-tagged EqtSM (*green*) and eGFP-tagged CHMP4B (*magenta*) and treated with 1 mM LLOMe for the indicated time. White arrows mark EqtSM-positive puncta that gradually accumulate CHMP4B. Scale bar, 5 μm. **b** Time-course plotting EqtSM- and CHMP4B-positive puncta normalized to maximum in cells treated as in (**a**). Data are means ± SD. *n* = 5 cells. **c** Fluorescence images of wild-type (WT) and SMS-KO HeLa cells treated with 1 mM LLOMe for the indicated time, fixed and then stained with DAPI (blue) and antibodies against CHMP4B (green) and actin (magenta). Scale bar, 10 μm. **d** Time-course plotting normalized CHMP4B intensity in cells treated as in (**c**). For each violin plot, the middle line denotes the median, and the top and bottom lines indicate the 75th and 25th percentile. In chronological order: *n* = 40 (WT), 40 (SMS-KO), 60 (WT), 60 (SMS-KO), 60 (WT), 60 (SMS-KO), 92 (WT), 70 (SMS-KO), 80 (WT), 79 (SMS-KO), 81 (WT), 87 (SMS-KO), 82 (WT), 78 (SMS-KO), 96 (WT), 101 (SMS-KO), 70 (WT) and 70 cells (SMS-KO) over two independent experiments. *P* values were calculated by unpaired two-tailed *t* test.

10% FBS (Pan Biotech; P40-47500). Murine RAW264.7 macrophages (ATCC TIB-71) were cultured in RPMI supplemented with 10% FBS. A HeLa cell line stably expressing CHMP4B-eGFP was kindly provided by Anthony Hyman (Max Planck Institute for Molecular Cell Biology and Genetics, Dresden, DE) and has been described previously[47]. HeLa CHMP3-KO cells have been described previously[28]. DNA transfections were performed using Lipofectamine 3000 (Thermo Fisher). Treatment with siRNAs (Qiagen) were carried out using Oligofectamine reagent (Invitrogen) according to the manufacturer's instructions. siRNA target sequences were: GFP, 5'-GCACCATCTTCTTCAAGGACG-3'; ALIX, 5'-CCUGGAUAAUGAUGAAGGA-3'; TSG101, 5'-CCUCCAGUCUU

CUCUCGUC-3'; nSMase-1, 5'-CAGCAGAGAGGUCGCCGUU-3'; nSMase-2, 5'-CAAGCGAGCAGCCACCAAA-3'.

**RT-qPCR**. To verify siRNA-mediated knock-down of gene expression, RNA was extracted from siRNA-treated cells using TRIzol reagent (Thermo Fisher Scientific). One microgram of RNA was used to synthesize cDNA with the Superscript III Reverse Transcriptase Kit (Thermo Fisher Scientific, 18080051) according to the manufacturer's instructions. Quantitative PCR reactions were performed on a C1000 Thermal Cycler with a CFX96 Real-Time System (Bio-Rad) using

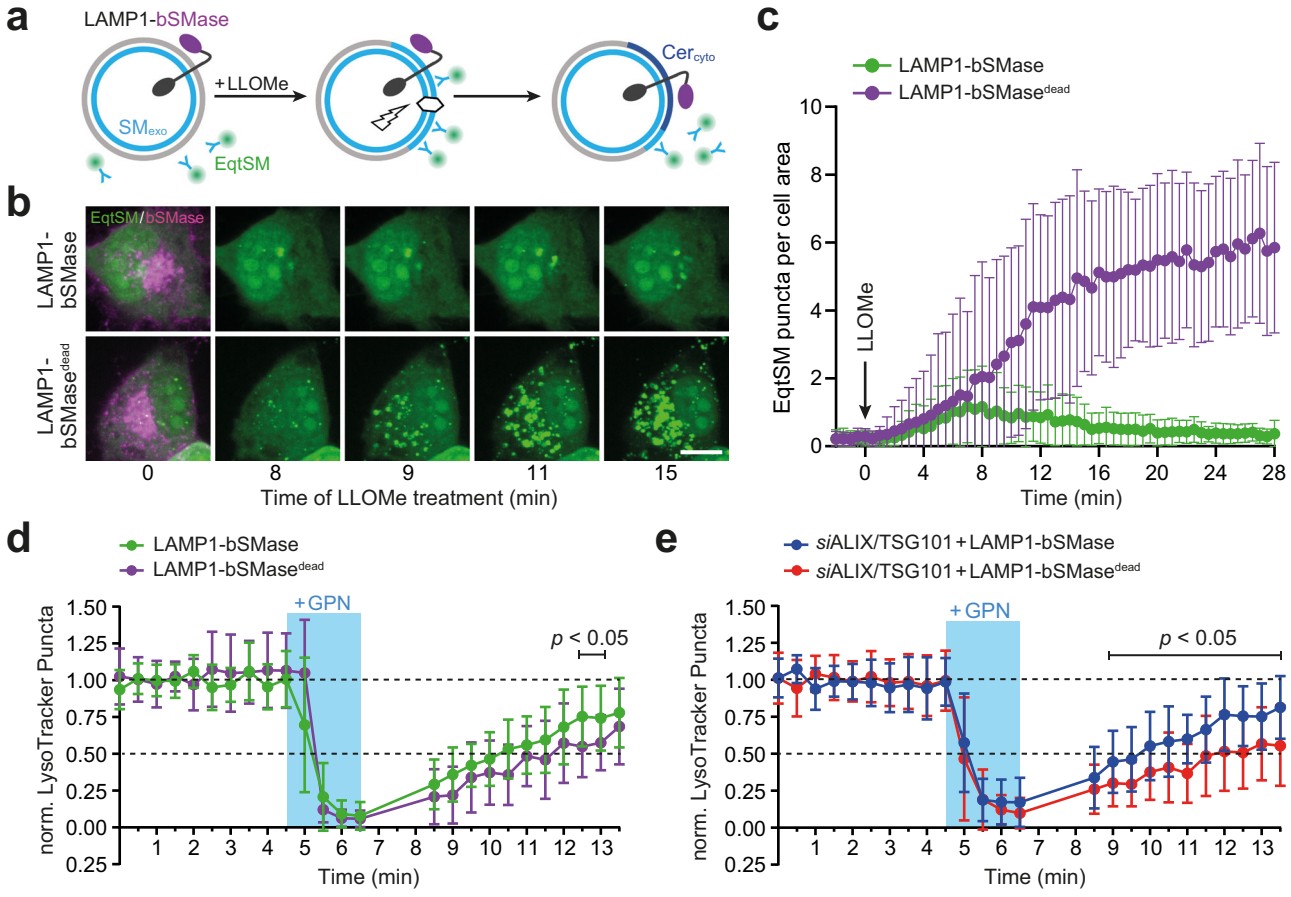

**Fig. 6 Hydrolysis of cytosolic SM promotes lysosomal repair in ESCRT-compromised cells. a** Bacterial SMase (bSMase, magenta) was fused to the *C*-terminus of lysosomal membrane protein LAMP1, enabling an efficient metabolic turnover of SM translocated to the cytosolic surface of LLOMe-damaged lysosomes. **b** Time-lapse images of HeLa cells co-expressing GFP/V5-tagged LAMP1-bSMase or LAMP1-bSMase$^{dead}$ (magenta) and mKate-tagged EqtSM (green) treated with 1 mM LLOMe for the indicated time. Scale bar, 10 μm. **c** Time-course plotting EqtSM-positive puncta per 100 μm$^2$ cell area in cells treated as in (**b**). Data are means ± SD. *n* = 14 cells per condition over two independent experiments. **d** Time-course plotting LysoTracker-positive puncta in HeLa cells co-expressing GFP/V5-tagged LAMP1-bSMase or LAMP1-bSMase$^{dead}$ and mKate-tagged EqtSM during and after a 2-min pulse of GPN, normalized to the initial number of puncta. Data are means ± SD. *n* = 16 cells for LAMP1-bSMase and 20 cells for LAMP1-bSMase$^{dead}$ over three independent experiments. *P* values were calculated by unpaired two-tailed *t* test. **e** Time-course plotting LysoTracker-positive puncta in HeLa cells pre-treated with siRNAs targeting ALIX/TSG101 (72 h) and expressing LAMP1-bSMase or LAMP1-bSMase$^{dead}$ during and after a 2-min pulse of GPN. Data are means ± SD. *n* = 23 cells for *si*ALIX/TSG101 + LAMP1-bSMase and 29 cells for *si*ALIX/TSG101 + LAMP1-bSMase$^{dead}$ over three independent experiments. *P* values were calculated by unpaired two-tailed *t* test.

Maxima™ SYBR™ Green/ROX 2x qPCR Master Mix (Thermo Fisher Scientific, K0221). Each reaction contained 400 ng of cDNA and 0.3 μM each of sense and antisense primers in a total volume of 10 μl. Initial denaturation was at 95 °C for 10 min. Cycles (*n* = 40) consisted of 10 s denaturation at 95 °C, 30 s annealing at 57 °C and 30 s of extension at 72 °C. Analysis of a single PCR product was confirmed by melt-curve analysis. All reactions were performed in triplicate. Expression data were normalized using actin as a reference. Ct values were converted to mean normalized expression using the 2$^{-ΔΔCt}$ method[48] and CFX Manager software version 2.1 (Bio-Rad Laboratories). Primers used were: nSMase-1-sense, 5′-G GTGCTCAACGCCTATGTG-3′; nSMase-1-antisense, 5′-CGTCTGCCTTCT TGGATGTG-3′; nSMase-2-sense, 5′-CAACAAGTGTAACGACGATGCC-3′; nSMase-2-antisense, 5′-CGATTCTTTGGTCCTGAGGTGT-3′; actin-sense, 5′-A TTGGCAATGAGCGGTTCC-3′; actin-antisense, 5′-GGTAGTTTCGTGG ATGCCACA-3′.

**Generation of TMEM16F-KO cells.** To knock out TMEM16F in HeLa cells, a mix of CRISPR/Cas9 constructs encoding three different TMEM16F-specific gRNAs and a GFP marker was obtained from Santa Cruz (sc-402736). The TMEM16F specific gRNA sequences were: A/sense, 5′-CAGCCTTTGGTACACTCAAC-3′; B/sense, 5′-GAATCTAACCTTATCTGTCA-3′; C/sense, 5′- AATAGTACTCACA AACTCCG-3′. At 24 h post transfection, GFP-positive single cells were sorted into 96-well plates using a SH800 Cell Sorter (Sony Biotechnology), expanded, and analyzed for TMEM16F expression by immunoblot analysis. In addition, loss of TMEM16F function was verified by analyzing ionomycin-treated cells for surface exposure of phosphatidylserine using Annexin V staining. To this end, wild-type

and TMEM16F-KO HeLa cells were detached using trypsin, taken up in DMEM containing 10% FBS, washed in PBS and resuspended in Annexin V Binding Buffer (Biolegend, no. 422201) and then incubated in the presence of 15 μM ionomycin or 0.1% (v/v) DMSO for 10 min at 37 °C in 5% CO$_2$. Next, APC-Annexin V (Biolegend, no. 640920; 5 μl in 100 μl Binding Buffer) and propidium iodide (5 μg/ml; Sigma-Aldrich, P4170) were added and cells were incubated for 10 min at RT. After the addition of 400 μl Annexin V Binding Buffer, cells were cooled on ice and then subjected to flow cytometry using a SH800 Cell Sorter (Sony Biotechnology). Flow cytometry data were analyzed using Sony Cell Sorter software version 2.1.5.

**Generation of nSMase-2-KO and SMS-KO cells.** To generate nSMase-2-KO and SMS1/SMS2 double-KO (SMS-KO) HeLa cells, a mix of CRISPR/Cas9 constructs encoding three different gRNAs per gene and the corresponding HDR plasmids were obtained from Santa Cruz (nSMase-2, sc401937; SMS1, sc-403382; SMS2, sc-405416). nSMase-2-specific gRNA sequences were: A/sense, 5′-CGTAGACCC CGACGTCGTAC-3′; B/sense, 5′-GAGTACATCCTGTACGACGT-3′; C/sense, 5′-GTGGCATTTGACGTCGTCTG-3′. SMS1-specific gRNA sequences were: A/sense, 5′-TGATACCACCAGAGTCGGCCG-3′; B/sense, 5′-TTGTACCTCGATCTT ACCAT-3′; C/sense, 5′-TAAGTGTTAGCATGACCGTG-3′. SMS2-specific gRNA sequences were: A/sense, 5′-TAACCGTGTGACCGCTGAAG-3′; B/sense, 5′-GG TCTTGCATAAGTGTTCGT-3′; C/sense, 5′-GTTACTACTCTACCTGTGCC-3′. Cells transfected with the nSMase-2-KO and SMS2-KO constructs were grown in medium containing 2 μg/ml puromycin at 48 h post transfection (Sigma-Aldrich, P8833). After 1–2 weeks, single drug-resistant colonies were picked, expanded, and analyzed for nSMase-2 and SMS2 expression by immunoblot analysis. A SMS1/2

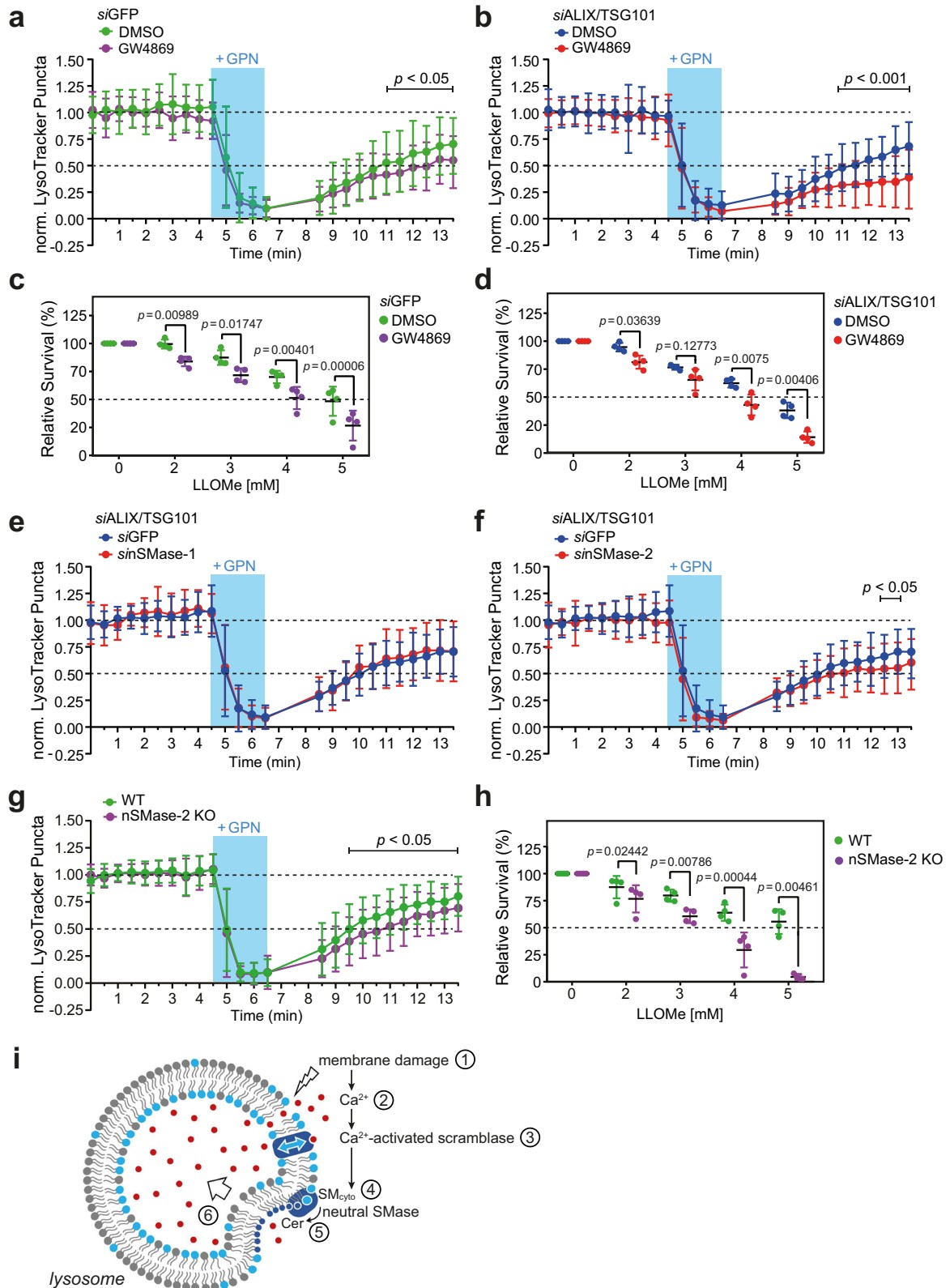

double-KO cell line (clone #11) was created by transfecting SMS2-KO clone #25 with SMS1-KO constructs as above, after ejection of the puromycin selectable marker using Cre vector (Santa Cruz, sc-418923) according to the manufacturer's instructions. Loss of SMS1 was confirmed by metabolic labeling of double-KO candidates with 4 μM of the clickable sphingosine (2S, 3R, 4E)-2-Amino-octadec-4-en-17-yne-1,3-diol (clickSph) in Opti-MEM (Fisher Scientific Scientific, 11520386) for 24 h. The synthesis of clickSph will be described elsewhere (S. Korneev and J. Holthuis, manuscript in preparation). Next, cells were washed in PBS, harvested,

and subjected to Bligh and Dyer lipid extraction[49]. Dried lipid films were click reacted with the fluorogenic dye 3-azido-7-hydroxycoumarin (Jena Bioscience, CLK-FA047) by addition of 64.5 μl of a freshly prepared click reaction mix containing 0.45 mM 3-azido-7-hydroxycoumarin and 1.4 mM Cu(I)tetra(acetonitrile) tetrafluoroborate in CH₃CN:EtOH (3:7, v-v) for 2.5 h at 45 °C without shaking. The reaction was quenched by the addition of 150 μl methanol, dried down in a Speed-Vac, dissolved in CHCl₃:methanol (2:1, v-v), and applied at 120 nl/s on a NANO-ADAMANT HP-TLC plate (Macherey-Nagel, Germany) with a CAMAG Linomat

**Fig. 7 Inhibition of neutral SMases disrupts repair of damaged lysosomes. a** Time-course plotting LysoTracker-positive puncta in HeLa cells pre-treated with siRNA targeting GFP (72 h) and GW4869 (10 μM) or DMSO (0.5%, 30 min) and pulse-treated with GPN. Data are means ± SD. $n = 52$ cells for DMSO and 39 cells for GW4869 over three independent experiments. **b** Time-course plotting LysoTracker-positive puncta in HeLa cells pre-treated with siRNAs targeting ALIX/TSG101 and GW4869 or DMSO and pulse-treated with GPN. Data are means ± SD. $n = 46$ cells for DMSO and 44 cells for GW4869 over three independent experiments. **c** Survival rate of HeLa cells pre-treated with siRNA targeting GFP after 5 h of exposure to LLOMe at indicated concentrations in the presence of GW4869 or DMSO. Data are means ± SD. $n = 3$ independent experiments. **d** Survival rate of HeLa cells pre-treated with siRNA targeting ALIX/TSG101 after 5 h exposure to LLOMe at indicated concentrations in the presence of GW4869 or DMSO. Data are means ± SD. $n = 3$ independent experiments. **e** Time-course plotting LysoTracker-positive puncta in HeLa cells pre-treated with siRNAs targeting ALIX/TSG101 and GFP or nSMase-1 and pulse-treated with GPN. Data are means ± SD. $n = 28$ cells for siGFP and 28 cells for sinSMase-1 over three independent experiments. **f** Time-course plotting LysoTracker-positive puncta in HeLa cells pre-treated with siRNAs targeting ALIX/TSG101 and GFP or nSMase-2 and pulse-treated with GPN. Data are means ± SD. $n = 28$ cells for siGFP and 40 cells for sinSMase-2 over three independent experiments. **g** Time-course plotting LysoTracker-positive puncta in WT or nSMase-2 KO HeLa cells pulse-treated with GPN. Data are means ± SD. $n = 58$ cells for WT and 66 cells for nSMase-2 KO over three independent experiments. **h** Survival rate of wild-type (WT) or nSMase-2 KO HeLa cells after 5 h exposure to LLOMe at the indicated concentration. Data are means ± SD. $n = 4$ independent experiments. $P$ values were calculated by paired (**c**, **d**, **h**) or unpaired two-tailed $t$ test (**a**, **b**, **e**, **f**, **g**). **i** Model illustrating how $Ca^{2+}$-activated SM scrambling and turnover may promote restoration of damaged lysosomes.

---

5 TLC sampler (CAMAG; Switzerland). The TLC plate was developed in CHCl₃:MeOH:H₂O:AcOH (65:25:4:1, v-v:v:v) using a CAMAG ADC2 automatic TLC developer (CAMAG; Switzerland). Fluorescent lipids were analyzed using a ChemiDoc XRS + with UV-transillumination and Image Lab 5.2 Software (Bio-Rad Laboratories).

**LC-MS/MS analysis**. Wild-type and SMS-KO HeLa cells were cultured for 24 h in Optimem. Next, cells were lysed using a BRANSON 250 Sonifier (12 ultrasound pulses; timer set ∞ ; duty cycle 30%; output 2). Aliquots of lysate containing 50 μg total protein were subjected to Bligh and Dyer lipid extraction ref. [49]. Prior to extraction, a standard mix containing phosphatidylethanolamine (PE 17:0/14:1) and ceramide (Cer 18:1/17:0) was spiked into each sample for normalization. Dried lipid extracts were dissolved in a 50:50 mixture of mobile phase A (60:40, water:acetonitrile, including 10 mM ammonium formate and 0.1% formic acid) and mobile phase B (88:10:2, 2-propanol:acetonitrile:H₂O, including 2 mM ammonium formate and 0.02% formic acid). HPLC analysis was via a C30 reverse-phase column (Thermo Acclaim C30, 2.1 × 250 mm, 3 μm, operated at 50 °C; Thermo Fisher Scientific) connected to an HP 1100 series HPLC system (Agilent) and a QExactivePLUS orbitrap mass spectrometer (Thermo Fisher Scientific) equipped with heated electrospray ionization (HESI) probe. LC-MS/MS analysis was performed as described previously[50]. Briefly, elution was performed with a gradient of 45 min. During the first 3 min, elution started with 40% of phase B and increased to 100% in a linear gradient over 23 min. 100% of B was maintained for 3 min. Afterward, solvent B was decreased to 40% and maintained for another 15 min for column re-equilibration. MS spectra of lipids were acquired in full-scan/data-dependent MS2 mode. The maximum injection time for full scans was 100 ms, with a target value of 3.000.000 at a resolution of 70.000 at $m/z$ 200 and a mass range of 200–2000 $m/z$ in both positive and negative mode. The 10 most intense ions from the survey scan were selected and fragmented with HCD with a normalized collision energy of 30. Target values for MS/MS were set at 100.000 with a maximum injection time of 50 ms at a resolution of 17.500 at $m/z$ 200. Peaks were analyzed using Lipid Search software version 4.1 (MKI, Tokyo, Japan). Peaks were defined through raw files, product ion, and precursor ion accurate masses. Candidate molecular species were identified by database (>1.000.000 entries) search of positive ($+H^+$; $+NH_4^+$) or negative ion adducts ($-H^-$; $+COOH^-$) with mass tolerance set at 5 ppm. Samples were aligned within a time window and results combined in a single report. From the intensities of lipid standards and lipid classes used, absolute values for each lipid in pmol/mg protein were calculated. Data are presented as pmol lipid per 100 pmol total phospholipid measured.

**Lentiviral transduction and lipid supplementation of SMS-KO cells**. HeLa SMS-KO cells with a stably integrated doxycycline-inducible SMS1 expression construct were created by lentiviral transduction. To this end, HEK293T cells were co-transfected with pInducer20-FLAG-SMS1 and the packaging vectors psPAX2 (Addgene, 12260) and pMD2.G (Addgene, 12259). The culture medium was changed 6 h post transfection. After 48 h, the lentivirus-containing medium was harvested, passed through a 0.45 μm filter, mixed 1:1 (v/v) with DMEM containing 8 μg/ml polybrene (Sigma-Aldrich, TR-1003) and used to infect HeLa SMS-KO cells. At 24 h post infection, the medium was replaced with DMEM containing 1 mg/ml G418 (Sigma-Aldrich, G8168), and the selective medium was changed daily. After 3–5 days, positively transduced cells were selected and analyzed for doxycycline-dependent expression of FLAG-SMS1 by immunoblot analysis, immunofluorescence microscopy, and metabolic labeling with clickSph as described above. For lipid supplementation studies, dried lipid films of egg phosphatidylcholine (egg PC; Avanti Polar Lipids, 131601) or a 70:30 mol% mixture of egg PC and egg SM (Sigma-Aldrich, S0756) were resuspended in distilled H₂O supplemented with 3% fatty acid-free BSA (Sigma-Aldrich, A6003) and sonicated in a water bath for 30 min to create 6 mM lipid suspensions. Next, lipid suspensions

were added dropwise to SMS-KO cells in Opti-MEM to a final concentration of 160 μM. After 16 h of incubation, cells were exposed to lysosome-damaging drugs and further processed as indicated.

**SMase activity assay**. HeLa cells were seeded in a six-well plate at 150.000 cells per well in DMEM supplemented with 10% FBS. After 24 h, cells were transfected with nSMase-2-V5, LAMP1-bSMase-V5-GFP or LAMP1-bSMase^dead-V5-GFP and grown for 24 h. Next, cells were harvested in ice-cold lysis buffer (25 mM Tris pH 7.4, 0.1 mM PMSF, 1× protease inhibitor cocktail), subjected to sonication (Branson Ultrasonic Sonifier), and centrifuged at 500×g for 10 min at 4 °C to obtain a post-nuclear fraction. Aliquots equivalent to 20 μg of total protein were included in a 100 μl of reaction mixture containing 50 mM Tris pH 7.4, 10 mM MgCl₂, 0.2% Triton X-100, 10 mM DTT, 50 μM phosphatidylserine (Sigma-Aldrich, P7769) and 50 μM C₆-NBD-SM (Biotium, 60031). Reactions were incubated at 37 °C for 2 h, terminated by addition of MeOH/CHCl₃, subjected to a Bligh and Dyer lipid extraction and then analyzed by TLC as described above.

**Cytotoxicity assay**. Cells treated with siRNAs were seeded in a 96-well plate (Greiner Bio-One, 655101) at 10.000 cells per well in DMEM supplemented with 10% FBS at 24 h after starting the treatment. After 24 h, the medium was replaced with Opti-MEM, and 24 h later LLOMe was added at the indicated concentration. After 3.5 h, PrestoBlue HS (Thermo Fisher Scientific, P50200) was added directly to the well to a final concentration of 10% (v/v) and incubated for 1.5 h at 37 °C. Next, absorbance at 570 nm was measured with 600 nm as reference wavelength using an Infinite 200 Pro M-Plex plate reader (Tecan Lifesciences). To calculate the relative percentage of survival, the measured value for each well (x) was subtracted by the minimum measured value (min) and divided by the subtrahend of the average measured value of untreated cells (untreated) and the minimum measured value (min); ((x-min)-(untreated-min)). To analyze the impact of GW4869 on LLOMe sensitivity, cells were seeded in a 96-well plate as above and grown for 48 h. Next, cells were treated with 10 μM GW4869 or 0.5% DMSO (vehicle control) for 30 min before LLOMe was added at the indicated concentration. Cell viability was assessed as described above.

**Time-lapse recording of laser wounding**. Laser wounding and time-lapse acquisition were performed using an Olympus model FV3000 laser scanning microscope (Olympus Europa SE & CO. KG) optically coupled to a fs laser system that comprises a regeneratively amplified fs laser (Pharos-HE-20; Light Conversion Inc.) and an optical parametric amplifier (OPA, Orpheus-Twins F; Light Conversion Inc.). The fs-pulses are adjusted collinear to the optical path of the continuous laser integrated in the microscope, enabling the simultaneous use of the continuous and pulsed laser as pumping source. Prior to the coupling into the microscope, neutral density filters can be inserted to tailor the average power and so the energy per pulse. The filter combination for simultaneous IR and blue emission was BP 488/730-1200. The wavelength of the pulsed laser was set at 900 nm, the repetition rate was 10 kHz, and the pulse duration was 180 fs, with a total exposure time of 3 s. The average power at the sample position was 900 μW, which implies that an energy per pulse of 90 nJ was achieved. For time-lapse recording of laser wounding, an UPLSAPO ×60 water immersed NA 1.2 objective with a custom infrared (IR) coating was used. Based on the diffraction-limited spot created by the objective, the photodamaged area was estimated to be ~500 nm. For laser wounding experiments, HeLa cells were seeded on 24 mm glass coverslips in a six-well plate at a density of 150.000 cells per well in DMEM containing 10% FBS. After 24 h, the medium was replaced with Opti-MEM and cells were transfected with EqtSM-GFP or EqtSol-GFP. After 24 h, Opti-MEM was replaced with Imaging Medium (IM; 30 mM HEPES, 140 mM NaCl, 2.5 mM KCl, 1 mM MgCl₂, 1.8 mM CaCl₂, 10 mM D-glucose, pH 7.4.) and cells were transferred to the

microscope. Time-lapse images were acquired every 5 sec before and after laser wounding (5 z-sections, 1 μm apart).

**Time-lapse recordings of cells exposed to organelle-damaging agents**. Time-lapse recordings of cells exposed to organelle-damaging drugs or pathogens were performed using a Zeiss Cell Observer Spinning Disc Confocal Microscope equipped with a TempModule S1 temperature control unit, a Yokogawa Spinning Disc CSU-X1a 5000 Unit, an ORCA Flash 4.0 V3 camera (Hamamatsu), a motorized xyz-stage PZ-2000 XYZ (Applied Scientific Instrumentation) and an Alpha Plan-Apochromat ×63 (NA 1.46) oil immersion objective. The following filter combinations were used: blue emission with BP 445/50, green emission with BP 525/50, orange emission BP 605/70. All images were acquired using Zeiss Zen 2012 acquisition software. At 48 h before imaging, cells were seeded into a μ-Slide eight-well glass-bottom chamber (Ibidi, 80827) at a density of 20.000 cells per well in DMEM supplemented with 10% FBS. After 24 h, the medium was replaced with Opti-MEM and cells were transfected with expression constructs encoding fluorescently-tagged proteins. After another 24 h, Opti-MEM was replaced with IM containing 30 mM HEPES, 140 mM NaCl, 2.5 mM KCl, 1 mM MgCl₂, 1.8 mM CaCl₂ and 10 mM D-glucose, pH 7.4. Next, cells were immediately transferred to the stage-top incubator preheated to 37 °C. The slide was allowed to equilibrate for 10 min before initiation of image acquisition. For experiments under Ca²⁺-depleted conditions, CaCl₂-free IM was used, which was supplemented with either 2 mM EGTA or 100 μM BAPTA-AM. A high concentration of BAPTA-AM was chosen because 25 μM had no effect. Cell viability and lysosome morphology/mobility were not affected during 60 min treatment at 100 μM BAPTA-AM. Time-lapse images were acquired every 10–30 s (six z-sections, 1 μm apart). After 2 min, organelle-damaging agents were added directly to the well without pausing image acquisition.

**M. marinum infection**. RAW264.7 cells were seeded into a SensoPlate™ 96-Well Glass-Bottom Plate (Greiner Bio-One, M4187) at a density of 10.000 cells per well in RPMI supplemented with 10% FBS. After 24 h, cells were transfected with EqtSM-GFP or EqtSol-GFP. At 24 h post transfection, cells were infected with M. marinum wild-type or ΔRD1 mutant strains constitutively expressing mCherry at an MOI of 10. Strains were kindly provided by Caroline Barisch (University of Osnabrück) and have been described previously[51]. The 96-well plate was centrifuged at 1250×g for 30 s and then incubated for 2 h at 37 °C. Next, cells were washed with PBS and fixed with 4% (w/v) paraformaldehyde (PFA) in PBS for 15 min at RT. For time-lapse imaging, cells were grown in phenol red-free RPMI medium supplemented with 30 mM HEPES and infected with the above M. marinum strains at an MOI of 25. After centrifugation of the 96-well plate at 1250×g for 30 s, cells were imaged using the Zeiss Cell Observer SD microscope set-up with images captured at 1 min time intervals (five z-sections, 1 μm apart).

**Salmonella infection**. For real-time imaging of Salmonella infection, HeLa cells were seeded into a μ-Slide eight-well glass-bottom chamber (Ibidi, 80827) at a density of 20.000 cells per well in DMEM supplemented with 10% FBS. After 24 h, the medium was replaced with Opti-MEM and cells were transfected with a Halo-tagged EqtSM expression construct. After 24 h, the cells were infected with 3.5 h (late logarithmic phase) subcultures of Salmonella enterica serovar typhimurium strain SL1344 harboring the dual-fluorescence reporter plasmid p4889[29] with a multiplicity of infection (MOI) of 50. Infection was synchronized by centrifugation for 5 min at 500×g followed by incubation for 25 min at 37 °C in an atmosphere of 5% CO₂ before extracellular bacteria were removed by washing thrice with PBS. Next, host cells were maintained in Opti-MEM containing 100 μg/ml gentamicin (AppliChem, A1492), labeled with 100 nM Janelia Fluor 646 HaloTag (Promega, GA1120) for 30 min, washed five times with PBS, and then imaged using the Zeiss Cell Observer SD microscope set-up with images captured at 2 min time intervals (five z-sections, 1 μm apart). For flow cytometric analysis of Salmonella infection, HeLa cells were seeded into surface-treated 12-well plates at a density of 4 × 10⁵ cells per well in DMEM supplemented with 10% FBS. After 24 h, medium was replaced with Opti-MEM and 24 h later cells were infected with S. enterica serovar typhimurium strain SL1344 harboring the dual-fluorescence reporter p4889 as above with a MOI of 15. Infection was synchronized by centrifugation for 5 min at 500×g followed by incubation for 25 min at 37 °C in an atmosphere of 5% CO₂ before extracellular bacteria were removed by washing with PBS. Next, host cells were incubated in Opti-MEM containing 100 μg/ml gentamicin (AppliChem, A1492) for 1 h. Afterward, cells were maintained in Opti-MEM with a reduced gentamicin concentration of 10 μg/ml for the rest of the experiment. At 3 h post infection, cells were washed twice with PBS and detached by incubation in 300 μl of Accutase (Sigma-Aldrich, A6964) for 10 min at 37 °C. Upon addition of an equal volume of PBS, cells were collected by centrifugation at 500×g for 10 min and then incubated in 250 μl Opti-MEM containing 200 μg/ml rifampicin (Sigma-Aldrich, R7382) and 200 μg/ml chloramphenicol (Sigma-Aldrich, C1919) to block bacterial transcription and translation but allow maturation of newly synthesized fluorescence proteins for 30 min at 37 °C. Samples were analyzed using an Attune NxT Cytometer (Thermo Fisher Scientific). At least 10.000 infected (dsRed-positive) cells were gated to calculate the proportion containing host cytosol-exposed Salmonella (dsRed- and sfGFP-positive). Experiments were performed in technical

triplicates and data were analyzed using Attune NxT Software version 4.2.0. Overview images were taken with Zeiss Cell Observer with LD A-Plan x 20 (NA 0.3) Ph1 Objective.

**Immunostaining of fixed cells**. For treatment with digitonin, HeLa cells were seeded onto 12-mm glass coverslips at a density of 40.000 cells per coverslip in DMEM supplemented with 10% FBS. After 24 h, the medium was replaced with Opti-MEM and cells were transfected with EqtSM-GFP. After another 24 h, cells were treated with 250 μM digitonin for 1 min, washed twice in Opti-MEM, incubated in Opti-MEM for 3 min at 37 °C and then fixed with 4% (w/v) PFA in PBS for 15 min at RT. After quenching in 50 mM ammonium chloride, cells were permeabilized using PBS containing 0.1% (w/v) saponin and 0.2% (w/v) BSA, immunostained for Na/K-ATPase and counterstained with DAPI. For treatment with LLOMe and ionomycin, HeLa cells were seeded onto glass coverslips and transfected with EqtSM-GFP as above. At 16 h post transfection, cells were treated with LLOMe (1 mM, 20 min) or ionomycin (5 μM, 20 min), washed three times with PBS, fixed, and then processed for immunostaining against LAMP1 and GFP as above. For monitoring recruitment of endogenous CHMP4B to LLOMe-damaged lysosomes, cells were seeded onto 12-mm glass coverslips at a density of 40.000 cells per coverslip in DMEM supplemented with 10% FBS. After 24 h, the medium was replaced with Opti-MEM. At 48 h post-seeding, cells were incubated with Opti-MEM containing 1 mM LLOMe for the indicated time period, washed with PBS, and then fixed with MeOH at −20 °C for 15 min. Next, cells were washed three times with PBS and permeabilized in PBS containing 0.3% (v/v) Triton X-100 and 1% (w/v) BSA for 15 min. Cells were immunostained for CHMP4B and actin, and counterstained with DAPI.

**Time-lapse recordings of LysoTracker-labeled cells**. At 72 h before imaging, cells were treated with siRNAs as indicated. At 48 h before imaging, cells were seeded in a μ-Slide eight-well glass-bottom chamber (Ibidi; 80827) at a density of 20.000 cells per well in DMEM supplemented with 10% FBS. After 24 h, the medium was replaced with Opti-MEM and cells were transfected with expression constructs encoding fluorescently-tagged proteins as indicated. At 24 h post transfection, Opti-MEM was replaced by IM containing 75 nM LysoTracker (LT) and the cells were immediately transferred to a stage-top incubator preheated to 37 °C. The slide was allowed to equilibrate for 10 min before initiation of image acquisition with the Zeiss Cell Observer SD microscope. Time-lapse images were acquired every 30 s (six z-sections, 1 μm apart). After 4.5 min of image acquisition, GPN was directly added into the well to a final concentration of 200 μM without pausing the acquisition. After 2 min of GPN exposure, acquisition was paused for 2 min to aspirate the GPN-containing medium, wash the cells once with LT-containing IM and add fresh LT-containing IM before acquisition was resumed. To analyze the impact of GW4869 on the recovery of LysoTracker fluorescence after transient GPN exposure, cells were seeded in a μ-Slide eight-well glass-bottom chamber, subjected to medium changes as above, and incubated for 48 h. Next, the cells were treated with 10 μM GW4869 or 0.5% DMSO (v/v, vehicle control) for 10 min in IM without LT and subsequently for 10 min in IM containing 75 nM LT. The GW4869 and DMSO concentrations were kept constant during and after the 2-min GPN exposure and images were acquired every 30 s as described above.

**Image analysis**. All image analyses were performed on the original, unmodified data with Fiji Image J2 software (version 2.3.0/1.53 f) and Image J Macros provided in Supplementary Information. Only cells that maintained a healthy morphology were included into the analysis. To quantify the number of EqtSM, Gal3 or CHMP4B-positive puncta during time-lapse imaging, the background was subtracted to remove noise and a manual threshold was set to exclusively include puncta above the signal of the cytosolic probe in untreated cells. Puncta with close proximity were separated using the watershed function. Next, for each time point, all puncta with pre-determined characteristics were counted automatically (size 0.2–5 μm², circularity 0.5-1). For cells co-expressing CHMP4B-eGFP and EqtSM-mKate, the nuclear area was excluded from the analysis. For normalization, the number of puncta for each time point was divided by the total measured area to account for size difference of cells and then multiplied by 100 to obtain the number of puncta per 100 μm² cell area. For normalization relative to the maximal value, the maximum number of puncta for each cell was determined and each time point was divided by the maximum value. To quantify the intensity of EqtSM-positive puncta upon laser damage, the images were first corrected for bleaching. Next, an ROI at the site of damage was selected and after a fixed threshold was implemented with the "Minimum" setting, the relative intensity in the ROI was measured for each time point. For quantifying LT-positive puncta, the background was subtracted to remove noise and an automatic threshold was set for $t = 0$ min. Puncta with close proximity were separated using the watershed function. Next, for each time point, all puncta with pre-determined characteristics were counted automatically (size 0.2–5 μm², circularity 0.5–1). For quantification, the average of the first five time points (0–2 min) was calculated and every time point was divided by the average. To quantify the LT accumulation efficiency, the background was subtracted to remove noise and an automatic threshold was set for $t = 10$ min. Puncta with close proximity were separated and for each time point, puncta with pre-determined characteristics were counted automatically (size 0.2–5 μm²,

circularity 0.5–1). For quantification, the average for the time points from $t = 8$ to $t = 10$ min was calculated and every time point was divided by the average. To quantify the intensity of the CHMP4B immunostaining in LLOMe-treated cells, the cell outline marked by actin immunostaining was used to measure the cell area. For the CHMP4B immunostaining, a pre-determined, fixed threshold was applied and the intensity above the threshold was measured. The intensity above the threshold was divided by the cell area. The average value for 10 min LLOMe treatment was set to 1 and all data points were divided by the average value. Each individual measurement was plotted in a violin plot.

**Statistics and reproducibility.** Except for some supportive experiments for which the outcome was clear cut and verified by complementary approaches (i.e., Supplementary Figs. 2b, c, 5, 6b, 8a, and 10c), each experiment was repeated at least once with similar results, using independent experimental samples and statistical tests as specified in the figure legends. Source data with sample sizes, number of technical and/or biological replicates, means, standard deviations, and calculated $P$ values (where applicable) are provided in the Source Data file for Figs. 1b, d, h, f, 2b, d, f, g, 3c, e, g, i, j–l, 4d, 5b, d, 6c–e, 7a–h, and Supplementary Figs. 1b, 2d, 3b, 4c, 7b, c, 9, and 11.

**Reporting summary.** Further information on research design is available in the Nature Research Reporting Summary linked to this article.

## Data availability
All data generated or analyzed in this study are included in the manuscript and supporting files. Uncropped scans of immunoblots, gels, and TLC plates are provided in Supplementary Information. Source data are provided with this paper.

## Code availability
All custom code is provided in Supplementary Information.

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

## Acknowledgements
We gratefully acknowledge Caroline Barisch, Anthony Hyman, and Yusuf Hannun for providing DNA constructs, cell lines, and bacterial strains, Florian Fröhlich and Stefan Walter for technical support in LC-MS, and Rainer Kurre for technical support in live-cell microscopy. This work was supported by the Deutsche Forschungsgemeinschaft (projects SFB944-P14 and HO3539/1-1 to J.C.M.H., SFB944-P4 and SPP-2225 HE1964/23-1 to M.H., and INST 901/179 to M.I.) as well as by the National Institute of General Medical Sciences of the United States National Institutes of Health (award R01GM095766 to C.G.B.).

## Author contributions
P.N. and J.C.M.H. designed the research and wrote the manuscript; P.N. performed the bulk of experiments and analyzed the results; F.S. carried out the *Salmonella* infection experiments, with critical input from M.H.; T.S. and A.H. generated and analyzed the SMS-KO, TMEM16F-KO, nSMase-2-KO, and *pInd*-SMS1 cell lines; L.V. and M.I. assisted with 2-photon laser damage; Y.K., Y.D., and C.G.B. designed and characterized the equinatoxin probes; E.S. performed EqtSM and LAMP1 co-localization experiments; C.J.C. provided intellectual expertise and helped to interpret experimental results. All authors discussed results and commented on the manuscript.

## Funding

## Competing interests
The authors declare no competing interests.
