## [Peer Review File · Nature Communications]

Ca²⁺-activated sphingomyelin scrambling and turnover mediate ESCRT-independent lysosomal repairEditorial Note: This manuscript has been previously reviewed at another journal that is not operating a transparent peer review scheme. This document only contains reviewer comments and rebuttal letters for versions considered at *Nature Communications*.

REVIEWER COMMENTS

Reviewer #4 (Remarks to the Author):

Niekamp and co-workers report a novel pathway for repair of damaged lysosomes, namely by calcium-mediated exposure of sphingomyelin on the cytosolic face of the lysosome membrane, followed by sphingomyelin hydrolysis mediated by neutral sphingomyelinase and (presumably) ceramide-mediated budding of the damaged area into the lysosome lumen. This is conceptually very interesting since it provides an alternative to ESCRT-mediated repair of the lysosome membrane. Overall the data are very well presented and make a convincing case for the model proposed by the authors.

I have not reviewed the previous versions of this manuscript but have seen the reviewer comments and the rebuttals from the authors. While I agree with the reviewers that it would have been interesting to know the identity of the scramblase involved and to see explicit evidence for the inward budding model, I think the quality and novelty of the revised manuscript make it well suited for publication in *Nature Communications*. I certainly disagree with reviewer #3 that this is a “descriptive” study. The authors have uncovered a novel molecular mechanism of significant biological importance, and I would like to recommend publication without further revision.

Reviewer #5 (Remarks to the Author):

Issues raised in previous rounds of review have largely been addressed, either with addition of data or discussion of why the question at hand seems out of scope for the present study. The study remains descriptive but most aspects of data interpretation are improved.

This reviewer, however, has a significant concern about concluding that the observed sphingomyelin scrambling is dependent on Ca^{2+} . This conclusion is based entirely on experiments carried out in cells treated with abnormally high concentrations of BAPTA-AM (100 μM), with no control to show that the effect is specific to chelating Ca^{2+} . Suitable controls could include using BAPTA-AM at lower concentrations, using the alternate chelator EGTA-AM, and using 5,5-difluoroBAPTA-AM with its reduced Ca^{2+} affinity to control for Ca^{2+} -independent effects of BAPTA. There are a variety of ‘off-target’ effects associated with high concentrations of BAPTA-AM (for example, inhibiting the Na, K-

ATPase PMID 29382785). This is a fundamental concern that should be addressed to ensure that the title of the paper accurately represents what can be concluded from the data as presented, and will have implications for searching for a putative Ca²⁺-activated lysosomal scramblase.

Reply to Reviewers' comments on NCOMMS-21-38839-T

Reviewer #4

Niekamp and co-workers report a novel pathway for repair of damaged lysosomes, namely by calcium-mediated exposure of sphingomyelin on the cytosolic face of the lysosome membrane, followed by sphingomyelin hydrolysis mediated by neutral sphingomyelinase and (presumably) ceramide-mediated budding of the damaged area into the lysosome lumen. This is conceptually very interesting since it provides an alternative to ESCRT-mediated repair of the lysosome membrane. Overall the data are very well presented and make a convincing case for the model proposed by the authors.

I have not reviewed the previous versions of this manuscript but have seen the reviewer comments and the rebuttals from the authors. While I agree with the reviewers that it would have been interesting to know the identity of the scramblase involved and to see explicit evidence for the inward budding model, I think the quality and novelty of the revised manuscript make it well suited for publication in Nature Communications. I certainly disagree with reviewer #3 that this is a "descriptive" study. The authors have uncovered a novel molecular mechanism of significant biological importance, and I would like to recommend publication without further revision.

We thank the reviewer for evaluating the previous rounds of reviews and for the positive feedback on our manuscript.

Reviewer #5

Issues raised in previous rounds of review have largely been addressed, either with addition of data or discussion of why the question at hand seems out of scope for the present study. The study remains descriptive but most aspects of data interpretation are improved.

This reviewer, however, has a significant concern about concluding that the observed sphingomyelin scrambling is dependent on Ca^{2+} . This conclusion is based entirely on experiments carried out in cells treated with abnormally high concentrations of BAPTA-AM (100 μ M), with no control to show that the effect is specific to chelating Ca^{2+} . Suitable controls could include using BAPTA-AM at lower concentrations, using the alternate chelator EGTA-AM, and using 5,5-difluoroBAPTA-AM with its reduced Ca^{2+} affinity to control for Ca^{2+} -independent effects of BAPTA. There are a variety of 'off-target' effects associated with high concentrations of BAPTA-AM (for example, inhibiting the Na, K-ATPase PMID 29382785). This is a fundamental concern that should be addressed to ensure that the title of the paper accurately represents what can be concluded from the data as presented, and will have implications for searching for a putative Ca^{2+} -activated lysosomal scramblase.

We thank the reviewer for evaluating the previous rounds of reviews and for the constructive feedback on our work. We acknowledge that the concentration of BAPTA-AM used for the experiments in Figure 2E and F (100 μ M) is high but this is not without precedent. Short exposure to high concentrations of BAPTA (200 μ M, 30 min) or prolonged exposure to medium concentrations (30 μ M, 30h) were found to be necessary to effectively clamp intracellular Ca^{2+} concentrations and acquire insights on the role of free cytosolic Ca^{2+} in neurons, muscle fibers and various other cell systems (e.g. PMID 10639101; PMID 31452956; PMID 26558774). We verified that cell viability and lysosome morphology/motility was not affected during 60 min exposure to 100 μ M BAPTA-AM. However, this does not rule out that BAPTA and its derivatives may exert effects that are partially independent of their Ca^{2+} binding activity. When added at a concentration of 25 μ M, BAPTA-AM did not have a significant impact on LLOMe-induced recruitment of EqtSM (data not shown). Because BAPTA is more selective for Ca^{2+} than EGTA and binds Ca^{2+} up to 400-times faster, using EGTA-AM as alternate Ca^{2+} chelator has limited value. A meaningful application of fluorinated BAPTA derivatives like 5,5-difluoroBAPTA-AM as 'negative' control for BAPTA-AM, as suggested by the reviewer, is not trivial and will require further experiments to ensure that these compounds are taken up by cells with the same efficiency (note that 5,5-difluoroBAPTA has ~4-fold reduced affinity for Ca^{2+} relative to BAPTA). Ultimate proof for the involvement of a Ca^{2+} -activated

scramblase in the cytosolic exposure of sphingomyelin on damaged lysosomes will require its identification. Our ongoing efforts are entirely focused on that task. Nevertheless, our current data unambiguously demonstrate that sphingomyelin scrambling in SLO-damaged or ionomycin-treated cells is strictly dependent on Ca^{2+} and abolished upon removal of TMEM16F, a Ca^{2+} -activated lipid scramblase located at the plasma membrane (Fig. 2A-D; Fig. S6). TMEM16F is part of a large protein family that comprises many poorly characterized members. Based on our present findings, we think it is fair to postulate that sphingomyelin scrambling in damaged lysosomes is catalyzed by a TMEM16F-related scramblase. In view of the above, we hope that Reviewer #5 acknowledges our reservations to embark on a next round of experiments that unlikely will provide conclusive proof.

REVIEWERS' COMMENTS

Reviewer #5 (Remarks to the Author):

I am satisfied with the reply provided, although I recommend including the fact that 100 μM BAPTA-AM was chosen because 25 μM had no effect in the text or legend. Beyond that, I agree with the authors that the next step - clearly beyond the scope of this MS - is to identify and characterize the proposed Ca^{2+} -activated lysosomal scramblase.

Reply to Reviewers' comments on NCOMMS-21-38839A

Reviewer #5

I am satisfied with the reply provided, although I recommend including the fact that 100 μ M BAPTA-AM was chosen because 25 μ M had no effect in the text or legend. Beyond that, I agree with the authors that the next step - clearly beyond the scope of this MS - is to identify and characterize the proposed Ca^{2+} -activated lysosomal scramblase.

We thank the reviewer for evaluating the previous rounds of reviews and for the constructive feedback on our work. As requested, we now added the following statement to the Methods section of the manuscript (p. 23, lines 562-565): "For experiments under Ca^{2+} -depleted conditions, CaCl_2 -free IM was used, which was supplemented with either 2 mM EGTA or 100 μ M BAPTA-AM. A high concentration of BAPTA-AM was chosen because 25 μ M had no effect. Cell viability and lysosome morphology/mobility were not affected during 60 min treatment at 100 μ M BAPTA-AM."